# DECOMPOSING LLM COMPUTATION WITH JETS

**Yihong Chen**$^{\delta}$, **Xiangxiang Xu**$^{\aleph}$, **Pontus Stenetorp**$^{\Upsilon}$, **Sebastian Riedel**$^{\Upsilon}$, **Luca Franceschi**$^{\Omega}$

$^{\delta}$OATML, University of Oxford, UK   $^{\Upsilon}$AI Centre, University College London, UK
$^{\Omega}$Independent researcher, Berlin, Germany   $^{\aleph}$University of Rochester, USA

`yihong.chen@cs.ox.ac.uk, luc.franceschi@proton.me,`
`xiangxiangxu@rochester.edu, {p.stenetorp,s.riedel}@cs.ucl.ac.uk`

## ABSTRACT

Large language models are becoming general knowledge engines for diverse applications. However, their computations are deeply entangled, resisting modularization which complicates interpretability, auditing, and long-term maintenance. We introduce JET EXPANSIONS, a framework for expanding recursive residual computational graphs using jet operators that generalize truncated Taylor series. Our method systematically decomposes language models into explicit input-to-output computational paths and complementary remainders. This *functional decomposition* provides a principled, knife-like operator for cutting through entanglement in LLMs, enabling scalable model inspection. We demonstrate how JET EXPANSIONS ground and subsume the popular interpretability technique *Logit Lens*, reveal an ensemble of an exponential number of paths analytically verify prior research, and support several interpretability applications, including sketching a transformer language model with $n$-gram statistics extracted from its computations and indexing model toxicity levels without curated benchmarks.

## 1 INTRODUCTION

Earlier artificial intelligence (AI) featured symbolic systems, representing knowledge in units, such as entities and relations. This mirrors how humans *might* conceptualize reality into *structures* for repeatable reasoning about the reality. In contrast, the latest advancement surfaces a *unstructured* paradigm, particularly since large language models (LLMs) trained with massive unorganized web texts (Radford et al., 2019; Brown et al., 2020; Touvron et al., 2023a). Both paradigms provide pathways to build knowledge engines that power diverse downstream applications (Chen, 2025).

Unlike symbolic methods, which format knowledge into explicitly addressable units customized for query calculation, LLMs distribute knowledge across billions of highly entangled parameters. This misalignment between *knowledge layout* and *computation layout* lies at the core of LLM opacity, raising regulatory concerns about transparency, auditability, and long-term maintainability. Once trained, LLMs cannot be easily inspected or updated. Knowledge operations that are trivial in symbolic AI, are far from straightforward in LLMs (Gehman et al., 2020; Carlini et al., 2021; Mitchell et al., 2021; 2022; Meng et al., 2022). This fuels growing demands for LLM interpretability, particularly in high-stakes domains such as healthcare (Smith, 2021; He et al., 2025; Comeau et al., 2025) and robotics (Wachter et al., 2017; Fernández-Becerra et al., 2024; Raptis et al., 2025).

Existing interpretability methods often take a *data-then-explanation* approach: curate inputs, hypothesize which sub-computations matter, and observe activations to refine the hypothesis (Wang et al., 2022; Meng et al., 2022; Goldowsky-Dill et al., 2023). We argue that the real challenge is structural: LLM computations are *entangled*, preventing us from isolating embedded knowledge into meaningful units. While one can gain valuable insights with data-driven interpretability approaches, we posit that the ability to *restructure* computation into smaller, less entangled, end-to-end components "systematically" – rather than "empirically" – is central to tackle such issues at scale.

We present JET EXPANSIONS, a principled, general-purpose framework for manipulating LLM computations. Noting that LLMs are particular types of residual networks (He et al., 2016; Vaswani et al., 2017), our key idea is to recursively expand residual computations using *jet operators* (Ehresmann, 1951), the functional counterpart of truncated Taylor series. This process yields functional

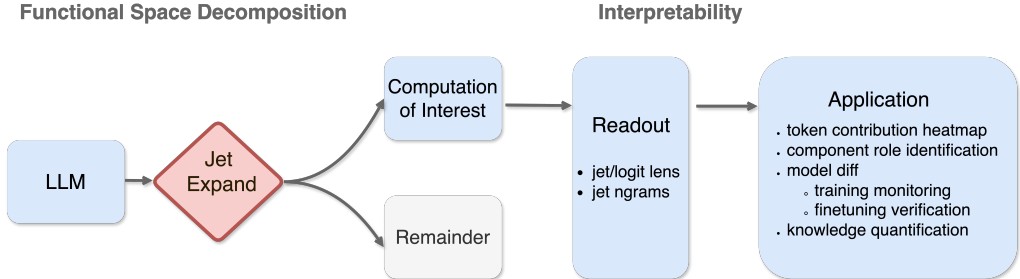

Figure 1: JET EXPANSIONS restructure residual computations into explicit input→output paths and a complementary remainder. From these paths we can extract logits, and $n$-grams without retraining or additional data. These readouts support downstream applications such as token contribution heatmaps, model comparison, training monitoring, and fine-tuning verification.

rewritings of the model into two parts: (i) explicit input→output polynomial functions, which we call *jet paths*, and (ii) complementary nonlinear remainders. Crucially, JET EXPANSIONS operates at a functional level, requiring no additional data, nor training. We show that JET EXPANSIONS encompass existing interpretability tools such as the Logit Lens (nostalgebraist, 2021b), and extend them to new instantiations such as extracting $n$-gram probability tables from LLMs. This enables dataset-free, symbolic sketches of transformer LLMs and allow us to perform global interpretability studies. Figure 1 illustrates the pipeline.

We validate our framework through case studies across several autoregressive LLMs (*GPT*, *Llama*, and *OLMo*). JET EXPANSIONS enable several empirical model inspection usages: i) understanding inner mechanisms via jet lens (Section 5.2.1); ii) assessing fine-tuning effects, e.g. quantifying toxicity levels with jet $n$-grams, showing RLHF alignment (Bai et al., 2022) reduces but does not eliminate toxic knowledge (Section 5.2.2); iii) analyzing pretraining dynamics, e.g. tracking how bigrams such as "at least" are promoted then suppressed in *OLMo* (App. I). These results demonstrate that JET EXPANSIONS provide a dataset-free operator for restructuring LLM computations, paving the way for more transparent, interpretable, and maintainable foundation models.

**Our contributions.**

1. A new angle on interpretability: treating it as *function decomposition*, rather than input-driven attribution or circuit identification on particular datasets.

2. A principled theoretical framework, based on jet operators, formally grounding existing tools such as *Logit Lens* (nostalgebraist, 2021b;a) and *path expansion* (Elhage et al., 2021).

3. Preliminary but wide-ranging case studies, revealing insights into LLM internal mechanisms, fine-tuning knowledge shifts, and toxicity levels.

## 2 BACKGROUND AND PRELIMINARIES

**Language models as residual networks.** We focus on transformer language models (Vaswani et al., 2017), which are residual networks (He et al., 2016) consisting of $L$ stacked residual blocks sandwiched between an encoder Enc and a decoder Dec. Formally, the full computation is

$$f : \mathcal{X} \to \mathcal{Y}, \qquad f = \text{Dec} \circ \left( \bigcirc_{\ell=1}^{L} (\text{id} + \gamma_\ell) \right) \circ \text{Enc}, \qquad (1)$$

where $\gamma_\ell$ is the non-linear transformation in block $\ell$. Unrolling the recursion, the hidden state after $\ell$ blocks is

$$h_\ell = h_0 + \sum_{j=1}^{\ell} \gamma_j \circ h_{j-1}, \qquad h_0 = \text{Enc}(z). \qquad (2)$$

This recursive form makes clear that residual links accumulate contributions from all preceding layers. We adopt the notion of *residual streams* (Elhage et al., 2021), where the computation of $h_l$ can be viewed as nested terms entangling contributions across blocks; see also (Veit et al., 2016). Table 1 summarizes the notation.

Table 1: Summary of notation used in the paper.

| Symbol | Meaning | Symbol | Meaning |
|--------|---------|--------|---------|
| $\mathcal{X}$ | Input space | $L$ | Depth (no. of blocks) |
| $V$ | Vocabulary size | id | Identity map |
| $\mathcal{Y} = \mathbb{R}^V$ | Output logits | $U$ | Unembedding matrix |
| $d$ | Hidden dimension | $\nu$ | Final normalization |
| $f : \mathcal{X} \to \mathcal{Y}$ | Full network | $h_\ell$ | Hidden state at layer $\ell$ |
| $\mathrm{Enc} : \mathcal{X} \to \mathbb{R}^d$ | Encoder | $\beta_\ell$ | Residual block at layer $\ell$ |
| $\mathrm{Dec} : \mathbb{R}^d \to \mathcal{Y}$ | Decoder | $\gamma_\ell$ | Residual transform inside block $\ell$ |
| $x_0$ | Base point (center) | $x$ | Variable |
| $D^j f(x_0)$ | $j$-th differential | $(x - x_0)^{\otimes j}$ | $j$-fold tensor product |
| $\mathrm{J}^k f(x_0)$ | $k$-jet at $x_0$ | $\mathrm{J}^k f$ | Jet operator |
| $P^k$ | Degree-$k$ polynomial space | $w_i$ | Jet weight for $i$-th base point |
| $\xi$ | Set of expanded terms | $\delta$ | Remainder of jet expansion |

**Taylor expansions and jets.**  To handle nonlinearities when restructuring residual computations, we turn to *jets* (Ehresmann, 1951), which generalize Taylor expansions. For $f \in C^{k+1}(\mathbb{R}^d, \mathbb{R}^d)$, Taylor's theorem at base point $x_0$ gives

$$f(x) = f(x_0) + \sum_{j=1}^k \frac{1}{j!} D^j f(x_0)\,(x - x_0)^{\otimes j} + O(\|x - x_0\|^{k+1}). \tag{3}$$

The $k$-th order *jet operator* abstracts this expansion as

$$\mathrm{J}^k f : \mathbb{R}^d \to P^k, \qquad \mathrm{J}^k f(x_0)(x) = f(x_0) + \sum_{j=1}^k \frac{1}{j!} D^j f(x_0)\,(x - x_0)^{\otimes j}, \tag{4}$$

or equivalently, by leaving the polynomial action implicit,

$$\mathrm{J}^k f(x_0) = f(x_0) + \sum_{j=1}^k \frac{1}{j!} D^j f(x_0).$$

Intuitively, $\mathrm{J}^k f(x_0)$ captures the local structure of $f$ up to order $k$, and we write $f(x) \approx_k \mathrm{J}^k f(x_0)(x)$ to indicate agreement up to order $k$. Jets thus provide a principled operator for rewriting computations of $f$ into decomposable pieces.

*Remark* 1 (Base points and variables as functions). When tracing back to the input $z \in \mathcal{X}$, base points $x_0$ and variables $x$ may themselves depend on $z$. In that case, jets define maps $\mathcal{X} \to \mathcal{Y}$ via $\mathrm{J}^k f(x_0(z))(x(z))$. For brevity, we often omit writing the variable $z$ explicitly when clear from context. See Appendix A for details.

## 3   RELATED WORK

**Mechanistic interpretability and path rewriting.**  A large body of work has sought to interpret the inner computations of large language models. One prominent category is *mechanistic interpretability*(MI) (Ferrando et al., 2024), which aims to reverse-engineer model computations by identifying, clustering, and labeling behaviors (Shah et al., 2024; Meng et al., 2022; Bricken et al., 2023) and attributing them to specific components, such as MLPs (Geva et al., 2021; 2022) or circuits (Conmy et al., 2023; Ferrando & Voita, 2024). However, these approaches often restrict analysis to atomic components (neurons, layers, or weights), which may not reveal the *full* mechanism of information processing. For example, Templeton et al. (2024) highlight the difficulty of drawing conclusions at the neuron level compared with higher-level feature representations, while Bolukbasi et al. (2021); Goldowsky-Dill et al. (2023) emphasize that many findings depend heavily on the chosen data distribution. A second category of approaches attempts explicit *path rewriting*. Veit et al. (2016) syntactically expand residual networks into exponentially many paths of varying length to study gradient behavior. Elhage et al. (2021) decompose one- and two-layer transformers into sums of uni-gram and bi-gram computation paths. Goldowsky-Dill et al. (2023) extend this line of work by developing path patching methods that aim to preserve functional faithfulness while isolating specific behaviors. Aligning with the second category, our approach manipulates functions directly rather than activations. It requires neither probe datasets (Belrose et al., 2023) nor sampling (Conmy et al., 2023; Ferrando & Voita, 2024; Voita et al., 2024). By allowing arbitrary portions of computation to be isolated from the monolithic transformer, JET EXPANSIONS abstract and generalize prior path-based characterizations (Veit et al., 2016; Elhage et al., 2021), where nonlinearities were often ignored or simplified (e.g. omitting layer norms, linearizing components, or implicitly assuming the nonlinear compositionality does not destroy the supposed independence of paths).

**$N$-gram models as symbolic counterparts.** $N$-gram models, dating back to Shannon (1948), represent one of the earliest symbolic approaches to language modeling. They store explicit probabilities of token sequences, e.g. $\Pr(w_i \mid w_{i-1}, \ldots, w_{i-n+1})$, in tabular form. This makes their *knowledge layout identical to their computation layout*: each symbol sequence has a directly addressable probability entry. Such symbolic modularity enabled early successes in language modeling (Goodman, 2001) and tasks like machine translation (Brants et al., 2007). While later work combined $n$-grams with networks (Liu et al., 2024), recent studies revisit their role in relation to LLMs: analyzing the ability of transformers to simulate $n$-gram statistics (Svete & Cotterell, 2024) or measuring agreement between LLM predictions and $n$-gram rulesets (Nguyen, 2024). This renewed attention motivates a direct bridge between $n$-grams and LLMs. JET EXPANSIONS provide this bridge, allowing corpus-free extraction of $n$-gram statistics *directly from LLMs* and thereby recovering a form of symbolic modularity within their entangled computations.

**Taylor expansions and jets.** Taylor expansions are ubiquitous tools in analyzing learning behaviours (Jastrzebski et al., 2017), notably with linearization ($k = 1$). For example, Belrose et al. (2024) applied Taylor expansion on the loss to demonstrate the learning preference of neural networks. Xu et al. (2022) used a second-order Taylor expansion over the data distribution to interpret optimal features. The generalized jet notions were introduced in machine learning in the context of automatic differentiation tools by Bettencourt et al. (2019), and is an experimental feature in Jax (Bradbury et al., 2018), but have been studied before (see e.g. Griewank & Walther, 2008). We leverage jets not merely as approximation tools, but as operators to restructure residual computations in LLMs into explicit input→output paths and complementary remainders.

# 4 RESTRUCTURING LLM COMPUTATION WITH JET EXPANSIONS

## 4.1 LINEAR CASE: EASY TO RESTRUCTURE

We begin with the linear case, where residual computations can be reorganized **_exactly_**. Assuming $\gamma_\ell(x) = A_\ell x$ for some $A_\ell \in \mathbb{R}^{d \times d}$, encoder $\text{Enc} = E$, and $\nu = \text{id}$, eq. (1) expands as follows

$$f = \text{Dec} \circ \left( \bigcirc_{\ell=1}^{L} (\text{id} + \gamma_\ell) \right) \circ \text{Enc} = U \left( \sum_{S \subseteq 2^{[L]}} \prod_{\ell \in S} A_\ell \right) E = \sum_{S \subseteq 2^{[L]}} f_S, \qquad (5)$$

where $2^{[L]}$ is the power set of $[L] = \{1, \ldots, L\}$ and each path $f_S = U \left( \prod_{\ell \in S} A_\ell \right) E = U W_S E$, with $W_\emptyset = I$, is itself a linear map from $\mathcal{X}$ to $\mathcal{Y}$. Thus, the entire network can be written as the sum of $2^L$ explicit input→output paths $f_S$. This exact decomposition makes linear residual networks intrinsically easy to restructure for interpretability because the output is a simple sum of its components: one can directly analyze the contribution of each path, study their interactions, and understand the global input→output relationships. In the nonlinear case, however, such a clean decomposition no longer holds, motivating the use of jets.

## 4.2 NONLINEAR CASE: JETS TO THE RESCUE

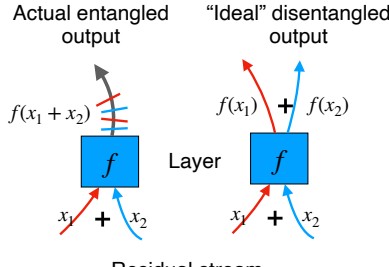

Figure 2: Convex combinations of jets disentangle a residual stream $h_\ell$ (a sum of terms) into sub-streams in function space, each isolated for further analysis.

$\mathrm{J}^k f(x_0)$ encodes all information about a function $f \in \mathcal{C}^{k+1}$ up to order-$k$ derivatives at a base point $x_0$, providing a vector-free representation of its local behavior. This makes jets a principled tool for reorganizing computations in LLMs. Lemma 1, proved in Appendix A, formalizes their *disentangling property*: a jet at a sum of inputs can be written as a convex combination of jets at individual inputs, up to higher-order error. This allows us to carve apart nested residual terms into separate, analyzable contributions (Figure 2).

**Lemma 1** (Disentanglement of Jets). *Let $f \in C^\infty(\mathbb{R}^d, \mathbb{R}^d)$, $k \in \mathbb{N}$, $N \in \mathbb{N}^+$, $\{x_i\}_{i=1}^N$ be a set of jet base points, and $w \in \triangle^{N-1} \subset \mathbb{R}^N$ be a set of jet weights (i.e., $w_i \geq 0$, $\sum_i w_i = 1$). Define the sum $\bar{x} = \sum_{i=1}^N x_i$ and $r = \max_i w_i \|x_i - \bar{x}\|$. Then the $k$-jet of $f$ at the sum $\bar{x}$ satisfies*

$$\mathrm{J}^k f \left( \sum_{i=1}^N x_i \right) = \sum_{i=1}^N w_i \, \mathrm{J}^k f(x_i) \; + \; O(r^{k+1}).$$

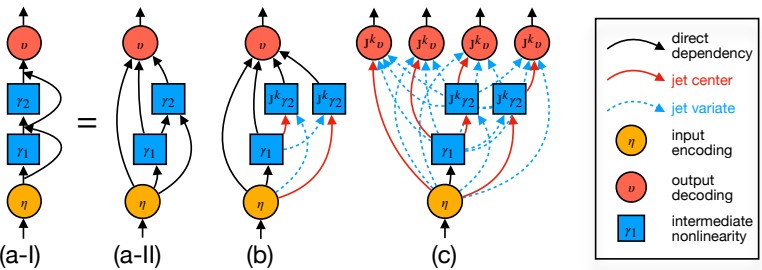

Figure 3: Carving a two-block net. (a) Nested entanglements. (b) Inner expansion at $\gamma_2$. (c) Outer expansion at Dec, yielding 4 explicit paths. Interpreting a model component entails selection. Jet centers trace paths through modules of interest; jet variates trace paths through those *excluded* from that focus.

*Example* 1 (JET EXPANSIONS of **ReLU**). Consider the **ReLU** activation function $\gamma : \mathbb{R} \to \mathbb{R}^+$ defined as $\gamma(x) = [x]_+$. For $x > 0$, $\gamma'(x) = 1$. For $x < 0$, $\gamma'(x) = 0$. Higher order derivatives are zero almost everywhere. If $x = x_1 + x_2$, then for almost every $x$, there exist $w \in \triangle^1$ such that

$$\gamma(x_1 + x_2) = w_1 \mathrm{J}^1\gamma(x_1)(x) + w_2 \mathrm{J}^1\gamma(x_2)(x) = w_1(\gamma(x_1) + \gamma'(x_1)x_2) + w_2(\gamma(x_2) + \gamma'(x_2)x_1).$$

In other words, for almost every base point $x = x_1 + x_2$, there exists a convex combinations of jets that can recover the original function $\gamma$ exactly. Indeed, if either $x_1, x_2 > 0$ or $x_1, x_2 < 0$, then any convex combination is exact. If only one of the two terms is positive, say $x_1 > 0$ and $x_2 < 0$, then we can set $w_1 = 1$ if $x_1 + x_2 \geq 0$ and $w_1 = 0$ otherwise ($w_2 = 1 - w_1$). The specular argument applies for the case $x_1 < 0$ and $x_2 > 0$. From a global perspective, we can think of jet weights $w_i = w_i(x_1, x_2)$ as optimizable functions of $x_1$ and $x_2$, rather then constants – and in the **ReLU** case, we obtain (almost everywhere) an exact first-order expansion. Conversely, one can see that the 0-th order JET EXPANSIONS of $\gamma$ is not globally exact.

### 4.2.1 MOTIVATING EXAMPLE: CARVING A TWO-BLOCK RESIDUAL NETWORK.

Now we consider how to use jets to carve a typical computation graph. We begin with the simplest nontrivial case: a network with two residual blocks. Using Equation (1), its full computation is

$$f = \mathrm{Dec} \circ \Big( \underbrace{\mathrm{Enc}}_{x_0} + \underbrace{\gamma_1 \circ \mathrm{Enc}}_{x_1} + \underbrace{\gamma_2 \circ \big( \mathrm{Enc} + \gamma_1 \circ \mathrm{Enc} \big)}_{x_2} \Big).$$

The nested parentheses entangle contributions: the outer (purple) grouping mixes everything, while the inner (orange) ties $\gamma_2$ to both $x_0$ and $x_1$. Traditional MI would select paths syntactically, akin to selecting modules in a PyTorch computation graph, ignoring these nesting effects. Jets let us cut both levels systematically and isolate the contributions of different input→output paths.

**Step 1: Inner expansion.** At $\gamma_2$, taking $\{x_0, x_1\}$ as jet base points and using Lemma 1, the residual stream $x_2 = \gamma_2 \circ (x_0 + x_1)$ can be decomposed as

$$x_2 \approx_k \mathrm{J}^k\gamma_2(x_0 + x_1) = \underbrace{w_0 \mathrm{J}^k\gamma_2(x_0)}_{x_{20}} + \underbrace{w_1 \mathrm{J}^k\gamma_2(x_1)}_{x_{21}} + O(r^{k+1}),$$

so the original entangled stream $x_2$ separates into two sub-streams, as illustrated in Figure 3(b).

**Step 2: Outer expansion.** At Dec, the jet base points are updated from $\{x_0, x_1, x_2\}$ to $\{x_0, x_1, x_{20}, x_{21}\}$ after previous expansion. Using Lemma 1 and jet algebra (Proposition 1), yields

$$f \approx_k \mathrm{J}^k\mathrm{Dec}(x_0 + x_1 + x_{20} + x_{21}) = \underbrace{\bar{w}_0 \mathrm{J}^k(\mathrm{Dec}\circ\mathrm{Enc})}_{f_\emptyset} + \underbrace{\bar{w}_1 \mathrm{J}^k(\mathrm{Dec}\circ\gamma_1\circ\mathrm{Enc})}_{f_{\{1\}}}$$

$$+ \underbrace{\bar{w}_2 \mathrm{J}^k(\mathrm{Dec}\circ(w_0 \mathrm{J}^k(\gamma_2\circ\mathrm{Enc})))}_{f_{\{2\}}} + \underbrace{\bar{w}_3 \mathrm{J}^k(\mathrm{Dec}\circ(w_1 \mathrm{J}^k\gamma_2(\gamma_1\circ\mathrm{Enc})))}_{f_{\{1,2\}}} + O(r^{k+1}) + O(\bar{r}^{k+1}).$$

corresponding to four distinct input→output paths $f_\emptyset, f_{\{1\}}, f_{\{2\}}, f_{\{1,2\}}$. This is shown in Figure 3(c). Each term aligns with what one might pick manually as a "path" in the network, but here it arises *systematically* from the JET EXPANSIONS.

The toy example above illustrates two key principles of our approach: recursive expansion of the nesting terms, and the use of disentangling property (Lemma 1) to isolate entangled contributions. In deeper networks with many blocks, however, manual expansion becomes infeasible. This motivates our general-purpose algorithmic framework, bringing such expansion to any depths.

## 4.3 GENERAL FRAMEWORK

Algorithm 1 describes the core operation of JET EXPANSIONS. At each block $l$, the algorithm applies Lemma 1 to a set of jet base points $\mathcal{C}$. For convenience, here $L + 1$ indicates the final decoder, Dec. The outputs are: (i) $\xi$, the set of polynomial terms (jet paths), where each term is the jet expansion centered at each $x_i \in \mathcal{C}$; and (ii) $\delta$, a nonlinear remainder, collecting error stemming from the Taylor expansion in Equation (3) and error from Lemma 1. A key feature is that jet base points can themselves be the outputs of earlier expansions. This enables recursive application of `jet_expand` throughout the network, unrolling the computation graph into end-to-end input→output paths. In particular, when we apply `jet_expand` at the final decoder layer $l = L + 1$, we obtain a functional rewriting of the model. Assume $(\xi_{L+1}, \delta_{L+1}) =$ `jet_expand`$(f, L + 1, \mathcal{C}, k)$ for some choice of jet base points $\mathcal{C}$ and order $k$, then

$$f(x) = \sum_{e \in \xi_{L+1}} e(x, w) + \delta_{L+1}(x, w), \quad (6)$$

where the jet weights $w \in \triangle^{N-1}$ can be manually specified or optimized. Hence, recursive applications of `jet_expand` yield a rewriting of the model as a sum of explicit paths, plus a complementary remainder. Each path is an atomic unit of computation from input to output, mirroring the original function, but with simpler, additive structures. This decomposition is purely algebraic and requires no extra data collection.

---

**Algorithm 1** `jet_expand`$(f, l, \mathcal{C}, k)$

**Require:** Net $f$ as in eq. (1); block index $\ell \in [L+1]$; jet base points $\mathcal{C} = \{x_i\}_{i=1}^N$; jet order $k \in \mathbb{N}$

**Ensure:** Expanded polynomial terms $\xi$ with weights $w$, and remainder $\delta$

1: **if** $\ell \leq L$ **then**

2:      **// residual block from $f$**
3:      $\gamma_\ell \leftarrow f.\texttt{block}(l)$

4:      **// residual block computation**
5:      $h_\ell \leftarrow h_{\ell-1} + \gamma_l(h_{\ell-1})$

6:      **// jet expansion at block $l$**
7:      $\xi \leftarrow \{\, w_i \mathrm{J}^k \gamma_\ell(x_i) \,\}_{i=1}^N$

8:      **// jet expansion at residual link**
9:      $\xi \leftarrow \xi \cup \{\, w_i \mathrm{J}^k \mathrm{id}(x_i) \,\}_{i=1}^N$

10:      **// calculate remainder**
11:      $\delta \leftarrow h_\ell - \sum_{e \in \xi} e$
12: **else**

13:      **// jet expansion at decoder**
14:      $\xi \leftarrow \{w_i \mathrm{J}^k f.\texttt{Dec}(x_i)\}_{i=1}^N$
15:      $\delta \leftarrow f.\texttt{Dec}(h_L) - \sum_{e \in \xi} e$

16: **return** $(\xi, \delta)$

---

*Remark* 2 (Jet weights). The jet weights $w$ can be fixed, for example as $w_i = 1/N$, or optimized to minimize the remainder at a given $x$, such as in logit space. When the decoder is linear, this optimization is efficient, as

$$\|U\delta_{L+1}(x, w)\|^2 = \|v(h_L(z)) - \sum_{e \in \xi_{L+1}} e(x, w)\|_{U^\top U}^2,$$

representing the squared distance between the expansion and the residual stream in $\mathbb{R}^d$, with the metric induced by $U$.

*Remark* 3 (Remainders). Remainders generally do not vanish with increasing $k$, as the base points are user-defined. For linear residual networks, however, $\delta = 0$ for all $k \geq 1$, showing that Algorithm 1 exactly recovers the rewrite in Equation (5) for any $w$. In light of Equation (6), jet expansions should be viewed as algebraic rewrites of computational graphs, intended to aid interpretation rather than to minimize approximation error. In our experiments, $\delta$ is often small, and the cosine similarity between expanded and original logits approaches 1 (Figure 4, bottom). See App.B for detailed discussion.

**Lemma 2.** *Residual nets with only* **ReLU** *nonlinearites admit exact first-order* JET EXPANSIONS.

**Runtime.** Evaluating $\xi$ and $\delta$ at $x \in \mathcal{X}$ requires computing $k$th-order jets at cost $O(|\mathcal{C}|(F + kB))$, where $F$ and $B$ denote forward and backward passes of $f$. In practice, higher-order jets can be computed efficiently via recurrence relations and automatic differentiation primitives such as Jacobian-vector products (JVPs) (Griewank & Walther, 2008; Bettencourt et al., 2019). App.D reports empirical runtime scaling of our implementation.

## 5 Applications of Jet Expansions

### 5.1 Theoretical applications

JET EXPANSIONS offer a principled framework that unifies and grounds existing techniques, such as *Logit Lens*, while enabling systematic derivations of new methods. Here we present several concrete instantiations of JET EXPANSIONS, and set the stage for the subsequent empirical studies.

**(Super-)exponential expansion.** Algorithm 2 extends our two-block example to arbitrary depth, producing $2^L$ paths with uniform jet weights. This mirrors Veit et al. (2016)'s exponential view of residual networks, but in an explicit and principled way. We defer the discussion on the connection between the expansion and the ensembling view on LLM computation to App.C. For $k \geq 1$, each polynomial term can be decomposed further by degree, isolating higher-order block interactions, hinting at a *superexponential* ensemble perspective which we leave as future work.

---

**Algorithm 2** `exp_jet_expand(f, k)`

**Require:** Net $f$ as in Equation (1); jet order $k \in \mathbb{N}$.
**Ensure:** Expanded terms $\xi$ (with uniform weights, $|\xi| = 2^L$) and remainder $\delta$.

1: **// initialize expansion**
2: $\xi \leftarrow \{f.\text{Enc}, \gamma_1 \circ f.\text{Enc}\}$
3: **for** $\ell = 2$ to $L + 1$ **do**
4: $\quad (\xi, \delta) \leftarrow \texttt{jet\_expand}(f, \ell, \xi, k)$

5: $\quad$ **// reweight terms uniformly**
6: $\quad \xi \leftarrow \{e(\cdot, 1/|\xi|) \mid e \in \xi\}$
7: $\delta \leftarrow f.\text{Dec}(h_L) - \sum_{e \in \xi} e$
8: **return** $(\xi, \delta)$

---

**Jet lens and logit lens.** The *logit lens* (nostalgebraist, 2021b; Geva et al., 2021; 2022; Merullo et al., 2023; Belrose et al., 2023) is a widely used mechanistic interpretability tool that applies the decoder to intermediate hidden states:

$$\text{LogitLens}_\ell(z) = U\nu(h_\ell(z)) = \text{Dec}(h_\ell(z)).$$

Aimed at highlighting the iterative refinement of the prediction across blocks, it is related to early exiting or early decoding in the context of conditional computation (see e.g. Panda et al., 2016; Elbayad et al., 2020; Geva et al., 2022). We can rewrite the logit lens with jet operator as follows

$$\text{LogitLens}_\ell(z) = \text{Dec}(h_\ell(z)) = \text{J}^0\text{Dec}(h_\ell(z))(h_L(z)) = \text{J}^0_{h_\ell(z)}\text{Dec}(h_L(z)).$$

Here we retain the argument $h_L(z)$ to emphasize that the zeroth-order jet is applied within the full computation, as if the jet operator acts like a knife: slicing the network at layer $\ell$ and replacing the sliced computation with a truncated jet expansion. Indeed, $\text{Dec}(x) \approx_{k=0} \text{J}^0_{h_\ell(z)}\text{Dec}(x) = \text{Dec}(h_\ell(z))$, so the *logit lens* coincides with the zeroth-order jet of the decoder at the base point $h_\ell(z)$, equivalently `jet_expand(f, L + 1, {h_\ell}, 0)`. This perspective suggests two direct generalizations of logit lens. First, *iterative jet lens* extends logit lens to higher-order jets: `jet_expand(f, L + 1, {h_\ell}, k), k \geq 1`. Second, *joint jet lens* expands with a broader set of base points rather than merely $\{h_\ell\}$: `jet_expand(f, L + 1, {γ_\ell ∘ h_{\ell-1}}_{\ell \in [L]}, k)`, thereby highlighting contributions of each block instead of the cumulative refinement of the residual stream.

**Jet $n$-grams.** $n$-gram statistics have gain traction in analyzing LLMs (Elhage et al., 2021; Svete & Cotterell, 2024; Nguyen, 2024), but existing methods rely on probing datasets. With JET EXPANSIONS, we can extract $n$-grams directly from the model. Concretely, since a model can be rewritten into a sum of polynomial terms or jet paths, we can select paths of interest, particularly shorter ones, and evaluate them over the entire vocabulary space or its cartesian product to record the resulting logits. Formally, given a model's expanded terms (jet paths) $\xi$, each path $e \in \xi$ defines a function $e : \mathcal{X} = V^{n-1} \to \mathbb{R}^V$. The $n$-gram score for token $i$ given a context $x$ is $s(x)[i] = \sum_{e \in \xi} e(x)[i]/|\xi|$. By evaluating $s(x)$ for all $x \in V^{n-1}$, we obtain a complete $n$-gram table $(x, i, s(x)[i]), x \in V^{n-1}, i \in V$, where $(x, i)$ identifies the $n$-gram and $s(x)[i]$ gives its score.

**From theory to applications.** Applying jet expansions in practice means choosing which portions of the computation of $f$ one is interested in carving out, and consequently, applying the expansion operator `jet_expand(f, ℓ, C, k)` as many times as needed, with appropriate layers $\ell$, centers $C$, and orders $k$, in order to obtain input-to-output functional, as remarked in eq. (6). After we specify a computation method for the jet weights, we obtain a set of functions (the jet paths) and a reminder.[1] All empirical objects in our experiments correspond to evaluations of such paths either on specific input sequences (e.g., "new simple neural architecture, the Transformer") or on the entire domain (vocabulary) of the resulting expansions. Following previous work, we use the first *input-specific* evaluation method for lenses experiments (section 5.2.1). We use the second *function-specific* evaluation method for all experiments concerning jet $n$-grams (section 5.2.2).

In particualr, we note that specific paths, like bi/tri-gram paths, restrict the effective domain to short sequences, where the domain $\mathcal{X} = V^{n-1}$ is treatable since $n \in \{2, 3\}$. This allows us to perform *exhaustive function-specific evaluations* and produce entire likelihood tables. For instance, evaluating a bi-gram path $e : V^1 \times [0, 1] \to \mathbb{R}^V$ at a weight $\bar{w}$ yields a complete table, $(x_i, x_j, e(x_i, \bar{w})[x_j])$ where $x_i, x_j \in V$ are tokens. These symbolic tables, which we dub *jet $n$-grams*, enable a holistic, dataset-free characterization of the model's global behavior, supporting global, dataset-free analyses. In our experiments, we implements such analysis via: (i) *top-K scoring bi-grams*, which reveal broad behavioral tendencies of the model, and (ii) *keyword-conditioned bi-gram mass*, which aggregates scores over bi-grams associated with a semantic category (e.g., toxicity), providing a scalar indicator of how much such knowledge is embedded in the model.

Hence, across both the input-specific (jet lenses) and function-specific (jet $n$-grams) settings, the connection to theory is direct: *each empirical quantity is simply a jet path evaluated on its natural domain*, with $(\ell, C, k)$ translating theoretical choices into concrete experimental readouts.

## 5.2 EMPIRICAL CASE STUDIES

**Setup.** We experimented with open-source LLMs including *GPT-2* (Radford et al., 2019), *GPT-Neo* Black et al. (2021), *Llama* (Touvron et al., 2023a;b; Rozière et al., 2024), and *OLMo* (Groeneveld et al., 2024). We run experiments on jet $n$-grams on 128-CPU servers with 1TB memory, using zero order jets and uniform jet weights. We run jet lenses experiments on a single laptop CPU, using various orders and optimizing jet weights with gradient descent. Further details about algorithms and evaluation metrics are given in App.E and App.F. Code is available at here.

### 5.2.1 CASE 1: ANALYZING LLM INNER WORKING

**Jet lens.** We use *jet lens* to analyze LLMs' mechanisms when processing individual examples. Figure 4 (top) visualize a joint jet lens for *GPT-Neo*-2.7B (Black et al., 2021). For space reason, other examples are deferred to App.G.1. The first row is the input sentence. The first column indicates the blocks. Here, a block, e.g. Block 1, contains one self-attention and one MLP module. All table cells depict top-1 tokens for the corresponding path through the particular block, following conventions from prior work (Belrose et al., 2023). We observe that the joint jet lens captures the synergy among different blocks, as the model prediction is decomposed into the contributions of several jet paths. Optimized jet weights are reported in the percentages.

In this sense, *jet lens* with $k > 0$ may serve as tools to systematically discover such synergic behaviors. We also find that higher-orders ($k > 0$) help iterative lenses deliver more meaningful interpretations than the *logit lens* ($k = 0$) for *GPT-Neo*-2.7B (see Figures 7 to 9). This is potentially due to their capability to trace indirect impacts of early layers on the final logits, which are otherwise missing under *logit lens*. Our findings are consistent with nostalgebraist (2021a); Cancedda (2024) where naive implementations of logit lens are shown to fail on *GPT-Neo* model family. Figure 4 (bottom) present mean cosine similarities of joint and iterative jet lenses with respect to model outputs for various *GPT* models and orders, averaged over 100 example sentences. The similarities are high and close to 1 for various $k$, showing however different behavior across model families and sizes. This indicates JET EXPANSIONS highly correlate with model outputs. In particular, the right plot compares the similarities of the logits obtained through iterative jet lenses for $k = 0$ (solid, line, the same as LogitLens) and for $k = 1$ (dashed lines), indicating an higher correlation of the latter with model outputs, potentially providing more faithful interpretations.

---

[1]In our experiments, we use either uniform or optimized weights, as described in remark 2.

|  | new | _simple | _neural | _architecture | , | _the | _Trans | former |
|---|---|---|---|---|---|---|---|---|
| Block 1 (7.36%) | , (3.40%) | ton (8.06%) | _network (8.57%) | _for (8.22%) | _which (7.51%) | _first (7.30%) | former (7.43%) | , (8.36%) |
| Block 2 (4.83%) | - (2.39%) | _ (5.23%) | _network (6.91%) | _for (4.98%) | _which (4.60%) | _neural (4.77%) | former (5.09%) | , (4.68%) |
| Block 4 (7.81%) | _impover (1.62%) | _unpop (1.29%) | _impover (1.31%) | _impover (1.28%) | _impover (1.25%) | _Neural (1.22%) | former (1.20%) | _Networks (1.32%) |
| Block 24 (6.02%) | , (5.74%) | _infographic (8.48%) | _network (8.76%) | _unve (8.45%) | _unve (7.67%) | _Neural (7.51%) | former (7.39%) | _model (8.45%) |
| Block 30 (6.24%) | _âĢ¦" (5.29%) | _ (1.31%) | _network (1.30%) | _for (1.29%) | _which (1.29%) | _neural (1.26%) | former (1.25%) | Âİ (1.31%) |
| Block 31 (7.76%) | !!" (5.33%) | _ (1.33%) | _network (1.31%) | _for (1.29%) | _the (1.26%) | _Conv (1.23%) | former (1.23%) | , (1.32%) |
| Block 32 (7.84%) | âĢ¦." (3.56%) | !?" (1.37%) | _network (1.36%) | , (1.33%) | _and (1.28%) | _neural (1.24%) | former (1.25%) | _model (1.32%) |

| Logits | _ | _ | _network | _for | _which | _neural | former | , |
|---|---|---|---|---|---|---|---|---|
| Expan. (0.993) | _ | _ | _network | _for | _which | _neural | former | , |

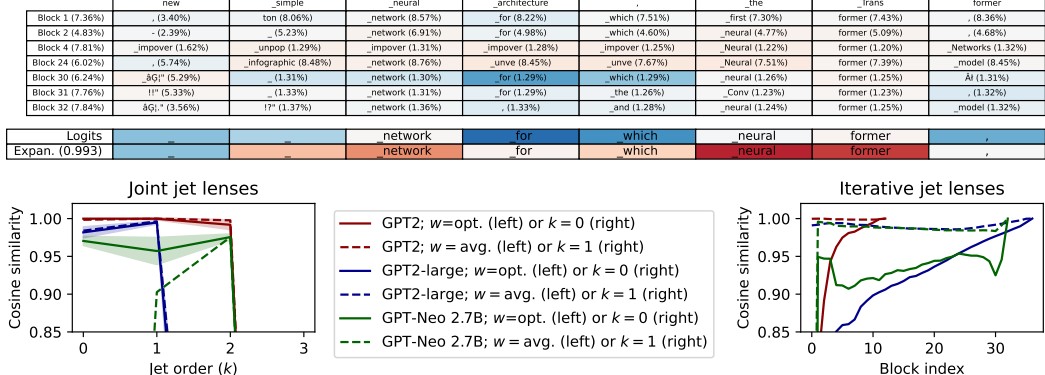

Figure 4: (**Top**) example of a joint jet lens on *GPT-Neo* 2.7B with $k = 1$, visualizing the seven blocks with highest average jet weights after optimization. Each table cell indicates the most likely token predicted by the jet path of each block. Optimized jet weight are displayed in the parenthesis next to the most likely token. We used a diverging blue-to-red color map tracking logit scores, centered at zero. The second table with two rows shows the model logits (Logits) and the expansion logits (Expan.), with cosine similarity (0.993) in parenthesis; in this case, all top-1 tokens perfectly coincide. (**Bottom**) plots of average cosine similarities between original and jet logits of joint (**left**) and iterative (**right**) lenses.

Table 2: MLPs in *OLMo*-7B and *Llama*-2-7B performing certain linguistic functions based on jet bi-grams extracted from the corresponding jet paths.

|  | *OLMo*-7B | | | | | *Llama*-2-7B | | | |
|---|---|---|---|---|---|---|---|---|---|
| **MLP Index** | 1 | 3 | 9 | 17 | 19 | 6 | 7 | 18 | 19 |
| **Linguistic Role** | -ly
-_else | -ing | -'t | -_than | -s | -ing | -es | -ing
-ity | -ly |
| **ΔLogit Intervened** | −4.19
−3.35 | −0.58 | −9.73 | −4.26 | −7.42 | −14.61 | −3.55 | −9.69
−11.93 | −9.14 |

**Jet paths of individual components.** By examining the representative jet bi-grams captured by jet paths of individual components, we can analyze their roles. In our case, we find some MLPs perform special linguistic functions. For example, in *OLMo*-7B, the jet path which passes through the 3rd MLP promotes the addition of the "-ing" suffixes to the current token. Similar MLPs with certain linguistic functions are listed in Table 2, where the negative Δlogit indicates removing the corresponding MLP harms the fulfillment of the particular linguistic functions. Note that the relationships between functions and components are not necessarily one-to-one mappings. Particularly paths through multiple MLPs might work together to complete one linguistic function e.g. MLP 6 and MLP 18 in *Llama*-2-7B can add "-ing" suffix. This echos work on circuit discovery (Elhage et al., 2022; Conmy et al., 2023; Ferrando & Voita, 2024), where the role of each component cannot be easily dissected and multiple components collaborate. Similar studies on the roles of attention heads can be be found in App.G.2.

### 5.2.2 CASE 2: ASSESSING FINE-TUNING EFFECT

Fine-tuning steers an LLM from pretraining's vastness towards focused, task-specific intent. These shifts ripple distributedly across high-dimensional parameter space, often escaping full capture without extensive benchmarking. Jet $n$-grams, however, render the changes legible directly from the weights, revealing model differences through their $n$-gram "diffing".

**Code fine-tuning.** Comparing *Llama*-2-7B with its code fine-tuned variants, *CodeLlama*, reveals that diffing jet bi-grams highlights code-specific patterns such as **kwargs or Assertion (Table 3), confirming the acquisition of programming knowledge. This suggests that jet bi-grams can serve as a practical tool to verify if fine-tuning effectively imparts knowledge in target domains.

**RLHF alignment.** While *ToxiGen* scores suggest detoxification of LLAMA-2-7B-CHAT, jet bi-gram masses remain nearly unchanged (Table 4), indicating toxic associations persist in latent form (additional evidence in App. H). Challenging prompts from RealToxicityPrompts (Gehman et al., 2020) confirm that these associations can still be triggered. Thus, RLHF appears to mask rather

Table 3: Bi-grams before and after code fine-tuning. For brevity, we show every 50th bi-gram among the top 1000. Bi-grams relevant to coding, such as **kwargs (a Python keyword), are highlighted. This demonstrates that our method can extract representative bi-grams reflecting fine-tuning quality.

| Rank | *Llama*-2-7B | *Codellama*-7B | *Codellama-Python*-7B |
|---|---|---|---|
| 0 | (_more, _than) | (_like, wise) | (_like, wise) |
| 50 | (_Now, here) | (_just, ification) | (_Like, wise) |
| 100 | (_system, atically) | (_in, _case) | (_all, udes) |
| 150 | (_all, erg) | (_get, ters) | (_no, isy) |
| 200 | (_on, ions) | (któber, s) | (output, ted) |
| 300 | (_other, world) | (_all, ud) | (Object, ive) |
| 350 | (_Just, ified) | (gebiet, s) | (_as, cii) |
| 400 | (_trust, ees) | (_Protest, s) | (_can, nab) |
| 450 | (_at, he) | (_deploy, ment) | (_transport, ation) |
| 500 | (_book, mark) | (Class, room) | (Tag, ging) |
| 550 | (_from, 而) | (_access, ory) | (_personal, ized) |
| 600 | (_WHEN, ever) | (_In, variant) | (_excess, ive) |
| 650 | (_where, about) | (_I, _am) | (_Add, itional) |
| 700 | (ag, ged) | (add, itionally) | (_**, kwargs) |
| 750 | (_he, he) | (_invalid, ate) | (name, plates) |
| 800 | (_all, anto) | (div, ision) | (_select, ive) |
| 850 | (_Tom, orrow) | (_process, ors) | (_Assert, ions) |
| 950 | (_Bach, elor) | (_set, up) | (_can, cellation) |

Table 4: Toxicity indexes for *Llama*-2-7B and *Llama*-2-7B-chat using different methods: *ToxiGen*, jet bi-grams, and *RealToxicityPrompts* challenge prompting. Higher numbers indicate higher toxicity scores on the corresponding benchmarks and higher toxic knowledge possession for jet bi-grams.

| | ToxiGen Score | Jet Bi-grams | RTP Challenging Prompts | | | |
|---|---|---|---|---|---|---|
| | Hartvigsen et al. (2022) | Mass of "toxic" bi-grams | No | Very mild | Medium | Hard |
| *Llama*-2-7B | 21.25 | 0.102 | 38% | 49% | 64% | 88% |
| *Llama*-2-7B-chat | 0.0 | 0.093 | 23% | 35% | 64% | 84% |

than erase toxic knowledge, a finding revealed directly by data-free jet bi-gram indices. This showcases a potential application of jet bi-grams in constructing *data-free* indices that reveal embedded knowledge, offering complimentary views beyond traditional data-driven benchmark evaluations.

# 6 CONCLUSION

We introduced jet expansions, a principled framework for decomposing the computation of transformer. Specialized to LLMs, our method systematically disentangles contributions of user-selected input→output paths from the overall computation, yielding interpretable functional components plus a complementary remainder. Operating directly in function space, jet expansions cut through entangled computation in recursive residual nets, and ground mechanistic interpretability techniques in approximation theory. This enables modular inspection: one can pull out paths of interest, e.g., logit lens, $n$-gram paths, while bracketing the rest as remainder.

**Limitations.** Jet expansions are not strict function approximations in the Taylor sense; they *rewrite* the computation into interpretable polynomial terms plus a remainder. Remainder sizes depend on the jet order $k$ and weight choices (hyperparameters), and expansions are not unique (higher orders contain lower orders). While graph manipulation is lightweight, systematic evaluation of many (and higher-order) paths can be costly; heuristics or subsampling may be needed for large input spaces. Our $n$-gram studies focused on bi- and tri-grams; longer-context expansions are left to future work.

**Implications and future work.** Moving beyond polynomial bases, we envision a Fourier-transform-style decomposition as a critical step toward *controllable* LLMs. For example, safety control by filtering out toxic "frequencies". We believe interesting future directions include connecting decompositions to attribution methods (e.g., the Shapley value), formalizing model equivalence via jet spaces to ground model diffing, and studying the implications of exponential ensembling path growth with model depth, hinted in algorithm 2. We see fruitful links to linear algebraic decompositions and to Markov/HMM viewpoints (e.g., structured decoding (Zhang et al., 2023)). Practically, beyond longer $n$-grams, we aim to develop dataset-free safety tools. Finally, although our experiments are mainly observational, jet expansions may help guide *interventions*, complementing causal tracing (Meng et al., 2022) and path patching (Goldowsky-Dill et al., 2023).

ACKNOWLEDGEMENTS

This work was partially funded under the Horizon Europe grant 101213369 DVPS. This work partially used CPUs from the DiRAC@Durham facility managed by the Institute for Computational Cosmology on behalf of the STFC DiRAC HPC Facility (www.dirac.ac.uk). The equipment was funded by BEIS capital funding via STFC capital grants ST/P002293/1, ST/R002371/1 and ST/S002502/1, Durham University and STFC operations grant ST/R000832/1. DiRAC is part of the National e-Infrastructure. Yihong thanks Lena, Karen, Javier, Nicola, William, Xinchi, Keenan, Hue, Yao, Jiayi, and Sohee for their valuable feedback on an earlier version of this work in 2024, as well as Yong, Yue, and the Tsinghua FIB Lab for hosting her research visit. Yihong is grateful to her family for their steady physical and emotional support throughout this project, to Bocan for her strength and creativity, and to Feier for her clarity and inspiration to revive this work when it was close to being set aside.

ETHICS STATEMENT

This work focuses on developing a mathematical framework (JET EXPANSIONS) for analyzing large language models. Our study does not involve human subjects, proprietary or sensitive data, or experiments that raise privacy, security, or legal concerns. We acknowledge that interpretability tools may potentially be misused to extract or expose harmful content (e.g., toxic or private knowledge) embedded in pretrained models. We use the public datasets for LLM toxicity research. We emphasize that our intent is to promote transparency, safety, and responsible analysis of LLMs, and we recommend future work carefully consider these implications in line with the ICLR Code of Ethics.

REPRODUCIBILITY STATEMENT

We have taken steps to ensure the reproducibility of our results. All definitions, assumptions, and theoretical proofs are included in the main text and appendix. Detailed algorithms (Algorithm 1, Algorithm 2) and mathematical derivations are provided for clarity. Experimental procedures, model families used (GPT-2, GPT-Neo, LLaMA, OLMo), and metrics are described in Section 5.2 and Appendix F. We will open-source the code implementing JET EXPANSIONS, extracting jet $n$-grams, and reproducing jet lenses, ensuring that all empirical results reported can be replicated.

LLM USAGE ACKNOWLEDGMENTS

We used LLMs to assist with grammar and writing polishing. All equations, analysis, and research contributions are our own.

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

# A    JETS AND EXPANSIONS

A jet of a function represents an equivalence class. We thus can perform algebraic operations among functional equivalence classes using jet algebra stated below.

**Proposition 1** (Jet algebra). *Let $f, g \in C^\infty(\mathbb{R}^d, \mathbb{R}^d)$ and $k \in \mathbb{N}^+$. Then,*

*(i)* $\mathrm{J}^k(af + bg)(x_0) = a\,\mathrm{J}^k(f)(x_0) + b\,\mathrm{J}^k(g)(x_0)$, *for $a, b \in \mathbb{R}$ (linearity);*

*(ii)* $\mathrm{J}^k f(x_0) \circ g \in \mathrm{J}^k f(x_0)$ *and* $\mathrm{J}^k f(x_0) \circ g(y) = \mathrm{J}^k f(x_0)(g(y))$ *(jet after endomorphisms);*

*(iii)* $g \circ \mathrm{J}^k f(x_0) = \{g \circ u\,:\, u \in \mathrm{J}^k f(x)\}$ *(endomorphism after jet);*

*(iv)* $\mathrm{J}^k(f \circ g)(x_0) = \mathrm{J}^k f(g(x)) \circ \mathrm{J}^k g(x_0)$ *(composition of jets);*

Properties *(i)-(iii)* follow directly from the definition; *(iv)* is a consequence of the chain rule and truncation. To reorganize residual computations typically used in LLMs, we rely on the disentangling property of jets, restated below.

**Lemma 1** (Disentanglement of Jets). *Let $f \in C^\infty(\mathbb{R}^d, \mathbb{R}^d)$, $k \in \mathbb{N}$, $N \in \mathbb{N}^+$, $\{x\}_{i=1}^N$ be a set of jet base points, and $w \in \triangle^{N-1} \subset \mathbb{R}^N$ be a set of jet weights (i.e., $w_i \geq 0$, $\sum_i w_i = 1$). Define the sum $\bar{x} = \sum_{i=1}^N x_i$ and $r = \max_i w_i \|x_i - \bar{x}\|$. Then the $k$-jet of $f$ at the sum $\bar{x}$ satisfies*

$$\mathrm{J}^k f\left(\sum_{i=1}^N x_i\right) = \sum_{i=1}^N w_i\,\mathrm{J}^k f(x_i) \; + \; O(r^{k+1}).$$

**Proof of Lemma 1**    Take $y \in \mathbb{R}^d$, $N \geq 1$, the set of jet base points $x_i \in \mathbb{R}^d$ for $i \in [N]$, jet weights $w \in \triangle^{N-1}$ and an order $k \geq 0$. Since $w$ belongs to the simplex $\triangle^{N-1}$, we have $\sum_{i=1}^N w_i = 1$. Multiplying $f(y)$ on both hands, we obtain

$$\sum_{i=1}^N w_i f(y) = f(y).$$

Applying eq. (3) (Taylor expansion) and the definition of jet with each $x_i$ as the center, the left hand side (LHS) becomes

$$\sum_{i=1}^N w_i f(y) = \sum_{i=1}^N w_i \left[ f(x_i) + \sum_{s=1}^k \mathrm{D}^s f(x_i)(y - x_i)^{\otimes s} + O(\|y - x_i\|^{k+1}) \right] \tag{7}$$

$$= \sum_{i=1}^N w_i \mathrm{J}^k f(x_i)(y) + O(w_i \|y - x_i\|^{k+1}), \tag{8}$$

At the same time, we can expand $f(y)$ at the right hand side (RHS) with $\sum_{i=1}^N x_i$ as the center

$$f(y) = \mathrm{J}^k f(\sum_{i=1}^N x_i)(y) + O(\|y - \sum x_i\|^{k+1}).$$

Now let us take $y = \sum_{i=1}^N x_i$ and observe that theremainder at RHS vanishes $O(\|y - \sum x_i\|^{k+1}) = 0$ and the remainder at LHS $O(w_i \|y - x_i\|^{k+1}) = O(w_i \|x_i - \sum_j x_j\|^{k+1})$. Finally we observe that the class of functions in the last $O$ are dominated by the class of function in $O(r^{k+1})$ where $r = \max_i \{w_i \|x_i - \sum_j x_j\|\}$ is the maximum remainder. This concludes the proof.

As a side note, jet weights would not need to form convex combinations, but rather linear combinations $\sum_i w_i = 1$. However, restricting to convex combinations has two major advantages:

- optimizing over a convex set guarantees the existence of maxima and minima (Weierstrass theorem) and uniqueness of minima if we are optimizing a strictly convex loss as in general is the case for expansions that only affect the decoder module.

- weights within the probability simplex have a clearer interpretation for interpretability purposes.

## B    REMAINDER SIZE OF JET EXPANSIONS

JET EXPANSIONS does not aim to provide approximation guarantees. Instead, in the following we clarify when small remainders are expected (input-specific cases) and when they are not (function-level expansions), but the expansions still yield meaningful, interpretable insights.

In ***input-specific*** evaluations (where we choose specific input sentences like "new simple neural architecture, the Transformer"), we expect reminders to be small, since we want to draw specific conclusions about how a model is behaving on a particular input. In these cases (i.e. in the Jet lenses experiments of §5.2.1) we do provide empirical studies, and we often find that the remainder $\delta$ is small. In order to measure the remainder between the expansion logit and the model logit, we compute the cosine similarity between them, since direct difference depends heavily on the input sentences (See App. F, "Cosine similarity as a remainder metric."). In short, we use cosine similarity as an indicator for checking if the remainder is small. Most of our heatmaps in the submission include this similarity measures in the parenthesis e.g. "Expan. (0.993)". We also compute average similarities over 100 examples and summarize the results in Figure 4 (bottom): expansion logits are frequently close to the model outputs, with cosine similarities in the 0.85–1 range, indicating that the extracted components capture most of the behavior for many concrete inputs i.e. smaller remainders. In general, the remainder size will depend on three factors:

1. the type of jet expansion one performs (e.g. iterative vs joint);
2. the non-linearity on which the jet is applied to (e.g. ReLu vs ELu, vs LayerNorm)
3. how far the variate is from the base point, which in turn fully depends on the chosen input sentence (see Equation (4)).

In ***function-level*** expansions (e.g., around the embedding function $\mathrm{Enc}$ for extracting jet bi-grams), remainders can be large and this is fully expected. Here, the goal is not approximation accuracy but to decompose the computation into interpretable paths. Concretely, let us consider bi-grams (§5.2.2): the model performs far more than the isolated mechanism of predicting a token based solely on the previous one. Therefore we do expect reminders between the model output and the extracted bi-gram paths to be *naturally* large. Yet the extracted paths remain *meaningful* precisely because they isolate a coherent part of the computation, even if they explain only a small fraction of the total behavior. A large remainder therefore does not invalidate the interpretation; it simply reflects that we are focusing on one specific, coherent part of the computation. This is a common situation across interpretability research, or in general when humans try to explain things. Conceptually, we find it useful to compare jet expansions to Fourier transform: even when the Fourier transform captures only part of the signal's spectrum and the resulting approximation error cannot be directly measured, the partial frequency information it provides can still be valuable for understanding the signal and diagnosing issues in the generating circuits.

## C    EXPONENTIAL EXPANSION AND ENSEMBLING PERSPECTIVE ON LLM COMPUTATION

Algorithm 2 recursively expands at all residual blocks. We see it as the maximal expansion (the upper limit of expansion) of JET EXPANSIONS. It aims to demonstrate that JET EXPANSIONS also provides theoretical grounds for Veit et al. (2016), whose analysis at the time lacked theoretical foundations and was presented heuristically and empirically, by manually selecting paths (i.e., $2^L$ gradient paths) for analyzing model behavior, where the exponentially many ensemble structures were syntactically noted rather than analytically derived.

With a closer look, we were surprised to find that several empirical procedures used in Veit et al. (2016) are exactly recoverable as specific instances of jet expansions. For example, the deletion of a module (Sec. 4.1 in Veit et al. (2016)) corresponds precisely to

$$\mathrm{jet\_expand}(f, l, h_{l-2}, 0),$$

i.e., a JET EXPANSIONS with the skip-layer upstream stream as the base point and with jet order $0$.

Another example is their gradient analysis. In the original exposition, the procedure is described operationally as: *"To sample a path of length $k$, we first feed a batch forward through the whole*

*network. During the backward pass, we randomly sample $k$ residual blocks. For those $k$ blocks, we only propagate through the residual module; for the remaining $n - k$ blocks, we only propagate through the skip connection.*" The outcome of these operational steps is equal to taking the derivative component of a first-order jet expansion over each block and evaluate it over a batch of inputs, as shown below. Note that the following reinterpretation focus on the gradients with respect to intermediate representations while empirically collecting gradients with respect to parametes is much easier.

**Path selection in backpropagation as first-order JET EXPANSIONS.** For a residual block,

$$\beta_l(h_{l-1}) = (\mathrm{id} + \gamma_l) \circ h_{l-1} = h_{l-1} + \gamma_l(h_{l-1}),$$

the derivative satisfies $D\beta_l[h_{l-1}] = I + D\gamma_l[h_{l-1}]$. Thus, propagating gradients through the full network $f = \beta_L \circ \beta_{L-1} \circ \cdots \circ \beta_1$ yields the Jacobian using the chain rule

$$Df(x_0) = \prod_{l=1}^{L} \big(I + D\gamma_l[x_{h-1}]\big).$$

On the other hand, the first-order jet of a residual block at $h_{l-1}$ is

$$\mathrm{J}^1(\beta_l)(h_{l-1}) = \big(\beta_l(h_{l-1}),\ I + D\gamma_l[h_{l-1}]\big),$$

where we use the pair notation for the jet, the first entry being the zero-th order term and the second entry being the first order term. Since jets compose according to

$$\mathrm{J}^1(f \circ g)(x) = \mathrm{J}^1 f\big(g(x)\big) \circ \mathrm{J}^1 g(x) \quad \text{(Proposition 1(iv)),}$$

we have the full first-order jet of $f$ as

$$\mathrm{J}^1(f)(x_0) = \mathrm{J}^1(\beta_L)(h_{L-1}) \circ \cdots \circ \mathrm{J}^1(\beta_1)(x_0),$$

Extracting the derivative component of this composite jet therefore yields the ordered product of the derivative factors from each block:

$$Df(x_0) = \big(I + D\gamma_L[h_{L-1}]\big) \circ \big(I + D\gamma_{L-1}[h_{L-2}]\big) \circ \cdots \circ \big(I + D\gamma_1[x_0]\big).$$

Composition of first-order jets multiplies the linear parts in this precise order:

$$Df(x_0) = \big(I + D\gamma_L[h_{L-1}]\big) \cdots \big(I + D\gamma_1[x_0]\big).$$

Expanding this product produces $2^L$ additive terms:

$$Df(x_0) = \sum_{\mathcal{S} \subseteq \{1,\ldots,L\}} \left( \prod_{l \in \mathcal{S}} D\gamma_l[h_{l-1}] \right),$$

each corresponding to a distinct choice of either $I$ or $D\gamma_l$ at every block. These terms match exactly, one-for-one, the "gradient paths" enumerated by Veit et al. (2016). Their operational procedure ("selecting" residual or skip gradients per block) amounts to selecting individual terms from this expansion. Consequently, their "gradient paths" are not independent computational input-output paths through the nonlinear network, but the combinatorial derivative components of the first-order jet (i.e., the first order term) of the residual network.

## D  RUNTIME OF JET EXPANSIONS

We report in fig. 5 a plot of the runtime for evaluating expansions originating from the joint jet lenses of section 5.2.1 as a ratio of the input model evaluation (forward pass), for both the uniform and the optimized jet weights $w$ setup, for different jet orders $k$.

## E  JET $n$-GRAMS AND THEIR ALGORITHMS

**General concept of $n$-gram models** The general concept of $n$-gram models linked to (transformer-based) language models involves defining or constructing mappings that functionally depend only on $n - 1$ input tokens (with the $n$-th token being the output token) to capture and describe the behaviour of the original language models. We are not the first to explore this idea; for instance Nguyen (2024) fits n-grams on the same dataset used to train the language models.

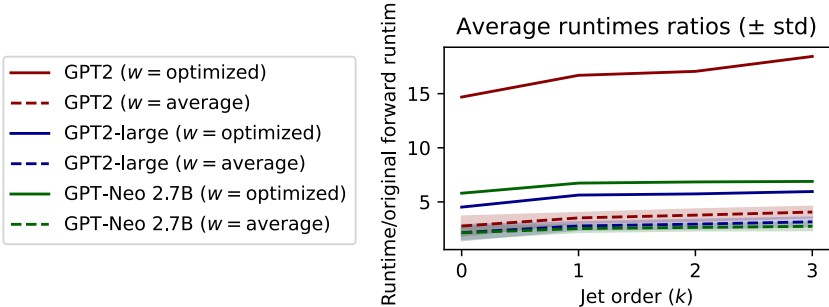

Figure 5: Empirical runtime of evaluations of JET EXPANSIONS originating form the joint jet lenses as a ratio of the evaluation of the input model.

**JET EXPANSIONS for in-model $n$-grams**    JET EXPANSIONS allow us to define $n$-grams statistics that are derived solely and directly from the model itself – producing *in-model* $n$-grams rather than *in-data* $n$-grams. This approach offers at least two significant advantages:

- **No repeated inference runs over prepared datasets:** It removes the need to prepare datasets for prompting LLMs, thereby avoiding repeated inference runs to collect activation patterns for interpretability analysis and reducing computational overhead. With a small $n$, direct expansion of LLMs into $n$-grams can be performed on CPUs, which are roughly an order of magnitude less expensive than GPUs.

- **Avoidance of fitting artifacts:** It avoids potential artifacts that could arise from the selection of external $n$-gram fitting methods.

We describe the detailed relationship between JET EXPANSIONS and bi-grams/tri-grams, which we used in our case studies. We will release code for these procedures and also provide equivalent algorithms that directly use transformer modules.

**Jet bi-grams**    Jet bi-grams are paths that do not pass through self-attention layers. In experiments, we focus on two types of bi-gram paths. a) the embedding-unembedding path that can be obtained as jet_expand($f, L, \{\mathrm{Enc}\}, 0$). b) paths that pass through one MLP module, assuming MLPs are at odd block indices in the residual network architecture, the procedure to extract the path is described from line 3 to line 7 in Algorithm 3. This procedure results in a series of functions in $\xi$, one for each MLP layer, that depend only on the last input token. Applying softmax normalization to their logit output allows these functions to define (conditional) bi-grams.

**Jet tri-grams**    Jet tri-grams involve paths that pass through at least one self-attention layer, with a need to isolate the contribution from the first token of the tri-gram. The procedure for extracting a 0-th order jet trigram path that passes through the $i$th self-attention layer (assuming it has one head) is described from line 1 to line 5 in Algorithm 4. This procedure yields a map that depends only on two input tokens, isolating the contribution of the $i$th self-attention layer on pairs of tokens. Once softmax normalization is applied, this defines a tri-gram. The tri-gram could represent either a skip trigram or a contiguous trigram, depending on how positional information is encoded (e.g., absolute positional embeddings versus rotary embeddings).

---

**Algorithm 3** Dataset-free extraction of jet bi-grams (MLP paths).

---

**Require:** Model $f$, total blocks $L$, vocabulary $V$

1: **// Initialize with encoder path**
2: $\mathcal{C} \leftarrow \{\text{Enc}\}$

3: **// Iterate over MLP blocks (odd indices)**
4: **for** $l = 1, 3, \ldots, L$ **do**
5: $\quad (\xi_l, \delta_l) \leftarrow \text{jet\_expand}(f, l, \{\text{Enc}\}, 0)$

6: $\quad$ **// Collect each expanded MLP term**
7: $\quad \mathcal{C} \leftarrow \mathcal{C} \cup \{e(\cdot, 1) \mid e \in \xi_l, e \neq \text{Enc}\}$

8: **// Expand over the decoder**
9: $(\xi, \delta) \leftarrow \text{jet\_expand}(f, L + 1, \mathcal{C}, 0)$

10: **// Evaluate over the input space**
11: **for all** $e \in \xi$ **do**
12: $\quad$ **for all** $x \in V$ **do**
13: $\quad\quad$ **for all** $i \in V$ **do**
14: $\quad\quad\quad$ Record $(x, i, e(x)[i])$ as the bi-gram score

15: **return** Symbolic bi-gram table $\{(x, i, e(x)[i])\}$

---

**Algorithm 4** Dataset-free extraction of jet tri-grams (Attention paths).

---

**Require:** Model $f$, total blocks $L$, target self-attention block index $l$, vocabulary $V$
1: $\mathcal{C} \leftarrow \{\text{Enc} \circ (x_{t-1}, x_t)\}$

2: **// Expand over the attention**
3: $(\xi, \delta) \leftarrow \text{jet\_expand}(f, l, \mathcal{C}, 0)$
4: $\mathcal{C} \leftarrow \{e(\cdot, 1) \mid e \in \xi_l, e \neq \text{Enc}\}$

5: **// Expand over the decoder**
6: $(\xi, \delta) \leftarrow \text{jet\_expand}(f, L + 1, \mathcal{C}, 0)$

7: **// Evaluate over the input space**
8: **for all** $e \in \xi$ **do**
9: $\quad$ **for all** $(x_{t-1}, x_t) \in V^2$ **do**
10: $\quad\quad$ **for all** $i \in V$ **do**
11: $\quad\quad\quad$ Record $(x_{t-1}, x_t, i, e(x_{t-1}, x_t)[i])$ as the tri-gram score

12: **return** Symbolic tri-gram table $\{(x_{t-1}, x_t, i, e(x_{t-1}, x_t)[i])\}$

---

**Practical computation vs. formal jet $n$-grams.** Algorithms 3 and 4 express bi-gram and tri-gram extraction in the jet-expansion formalism: we expand intermediate paths using `jet_expand`, collect the resulting functions into $\mathcal{C}$, and finally obtain a family of zeroth-order jet paths

$$e \in \xi, \qquad e : V^{n-1} \to \mathbb{R}^V,$$

where the arity of $e$ (one token for MLP paths, two tokens for single-attention paths) determines whether the path represents a bi-gram ($n = 2$) or a tri-gram ($n = 3$). Formally, an $n$-gram table is obtained by exhaustive evaluation of $e$ on its domain: recording $e(x)[i]$ for $x \in V$ in the bi-gram case, or $e(x_{t-1}, x_t)[i]$ for $(x_{t-1}, x_t) \in V^2$ in the tri-gram case. In practical implementations, these evaluations correspond to batched matrix computations over the entire vocabulary. For bi-grams, the embedding matrix $E \in \mathbb{R}^{|V| \times d}$ serves as the batch of all input tokens, and each MLP component represented in $\mathcal{C}$ acts on $E$ via matrix multiplications; after layer normalization and

projection through the unembedding matrix $U$, the resulting matrix has $(x, i)$ entry equal to $e(x)[i]$. For tri-grams, an analogous "pairwise embedding" tensor encodes all $(x_{t-1}, x_t)$ pairs at once, and the attention and subsequent linear operators act on this tensor in batch; projection through $U$ yields a three-dimensional array whose $(x_{t-1}, x_t, i)$ entry is exactly $e(x_{t-1}, x_t)[i]$. Thus, the formal jet description and the practical batched-matrix implementation are two views of the same operation: a jet path defines the atomic paths $\{e, e \in \xi\}$ which simplifies the large model, and the bi-gram table is obtained by evaluating $e$ simultaneously on all vocabulary elements using matrix multiplications.

## F   EXPERIMENTAL METRICS

In this section, we detail the measured quantities in each empirical case study.

**Cosine similarity as a remainder metric.**   In Section 5.2.1 (the lens experiments), we need a comparable metric for quantifying the size of the remainder term $\delta$ in the JET EXPANSIONS for a given input sentence. Let $m \in \mathbb{R}^V$ denote the model output logits over the vocabulary $V$, produced by a full forward pass, and let $e \in \mathbb{R}^V$ denote the logits predicted by the truncated jet expansion. A naïve approach would define the remainder as the difference vector $r = m - e$ and measure its magnitude using $\|r\|$. However, this direct norm-based measurement is highly sensitive to input-specific variation, such as sequence length, and the internal activation scaling of the model. As a result, the magnitude $\|m - e\|$ can be dominated by variations in logit norm rather than reflecting the intrinsic approximation error introduced by JET EXPANSIONS. To address this, we adopt cosine similarity as a scale-invariant measure of alignment between the model logits and their JET EXPANSIONS approximation. Formally, we compute

$$\cos(e, m) \;=\; \frac{e \cdot m}{\|e\|\|m\|},$$

where $\cdot$ denotes the standard dot product. A cosine similarity of $1$ means that JET EXPANSIONS preserves the model logits structurally. Conversely, lower cosine similarity values correspond to a larger remainder term in the expansion. Practically, cosine similarity enables us to disentangle remainder size from input-dependent logit scaling, offering a more interpretable and stable measure of the fidelity of JET EXPANSIONS. We therefore report cosine similarity throughout our experiments as our primary metric for assessing remainder size.

**$\Delta$ Logit after intervention.**   In Section 5.2.1, to compute $\Delta$ logits, we calculate the logits for the given $n$-gram both before and after applying the intervention, then determine the change in the logits. For example, consider the trigram (Lemma, let, s). We compute the logit of "s" conditioned on the input "Lemma let". The intervention involves removing the corresponding attention head (e.g., head 2). We then measure and report the change in the logit for "s" as a result of this intervention.

**Jet bi-gram comparison for code fine-tuning.**   In Section 5.2.2, we derive the top 1000 bigrams using Algorithm 3. These bigrams are then saved, for example, as CSV files, enabling the inspection and comparison of models via their respective bi-grams. This approach allows us to bypass the challenges of comparing models in the high-dimensional parameter space, where measuring behavioral-level differences can be difficult. We have developed a web UI demonstration where users can perform "model diffs" using the respective jet bi-grams. For example, Figure 6 demonstrates how this UI can be used to compare the base Llama-2-7B model with its coding fine-tuned versions.

**Jet bi-gram toxic mass.**   In Section 5.2.2, we introduce a method to quantify the possession of toxic knowledge. We compute jet bigram probability scores and calculate the cumulative conditional probability mass over a curated set of toxic bigrams, pairs of tokens specifically linked to toxic meanings in a predefined word list. The toxic mass ($M$) is formally defined as the sum of these conditional probabilities across the query set ($Q$):

$$M = \sum_{z \in Q} P(z_2 | z_1)$$

Here, $Q$ represents the query set comprising the toxic bigrams derived from the word list. In this way, we can measure model toxicity using simple query words instead of relying on extensive, curated prompting datasets.

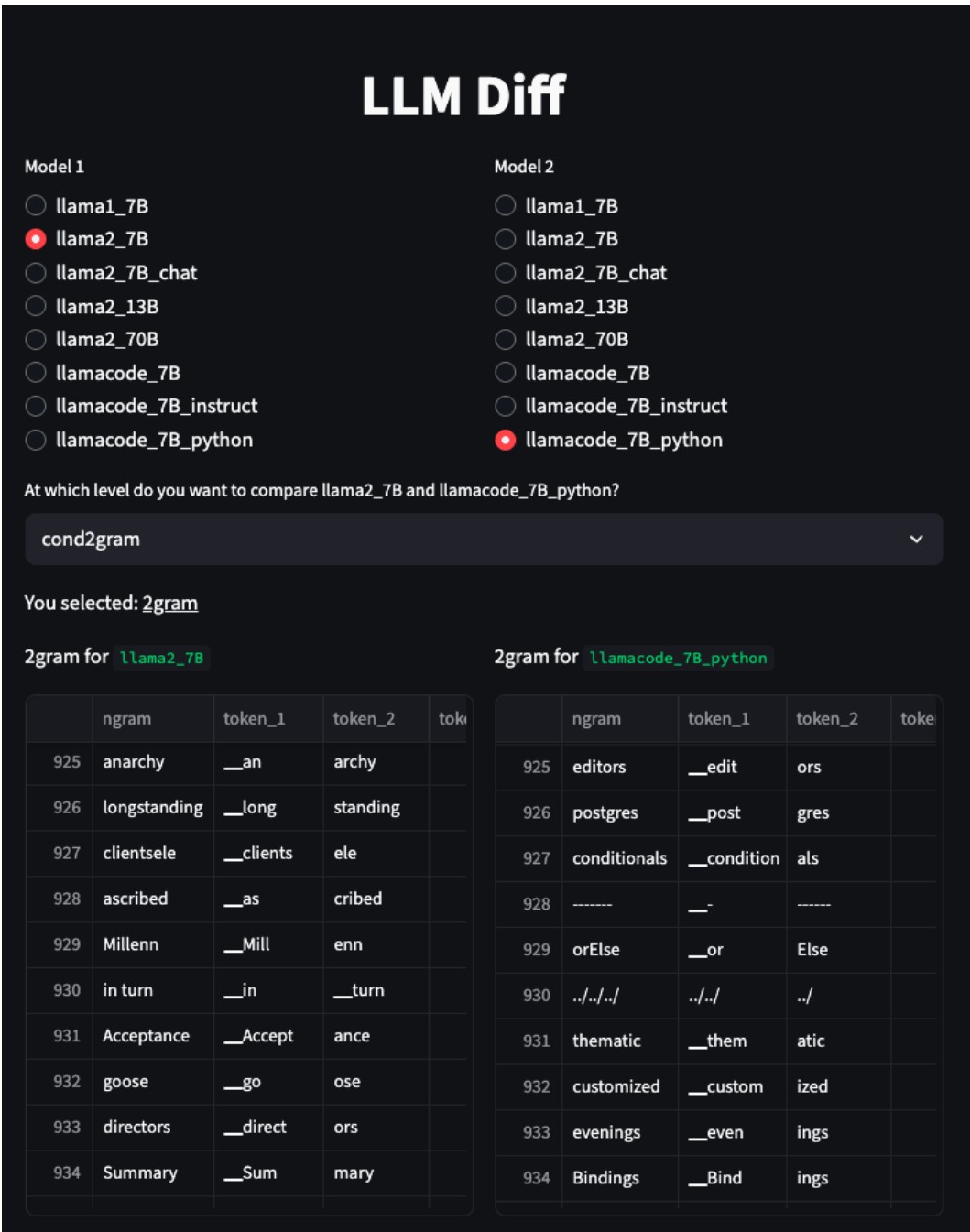

Figure 6: A web UI for running LLM Diff with jet n-grams.

**One-to-one bi-grams like and many-to-many bi-grams.** In Section I, we analyze the pretraining dynamics by checking the learning speed of bi-grams from different categories. One-to-one bi-grams are (approximately) unimodal bi-grams that concentrate all mass on a single token: i.e. given $z_1$, $\mathbb{P}_{\mathcal{D}}(z_2|z_1) \approx 1$ and given $z_2$, $\mathbb{P}_{\mathcal{D}}(z_1|z_2) \approx 1$ for a specific pair of token and close to 0 for all others. In the example in the paper, $z_1 =$ "&", and $z_2 =$ "amp". $\mathbb{P}_{\mathcal{D}}$ is the probability distribution induced by the pre-training data. Many-to-many bi-grams we refer to the opposite scenario where both the conditional probabilities are highly multi-modal. In the example $z_1 =$ "make" and $z_2 =$ "sure" we have that many other tokens can succeed $z_1 =$"make" or precede $z_2 =$"sure".

**Hit ratios of bi-grams.** The Hit Ratio (HR@n), often referred to as hit rate, is a metric commonly used in ranking tasks. In our context of Section I, we treat each checkpoint of the language model as a "ranker" of bi-grams. The Hit Ratio measures how effectively the current model checkpoint retrieves high-quality bi-grams from the set of all possible bi-grams. To quantify the model's progress, we define the bi-grams at the final step as the "good" bi-grams and measure how quickly the model approaches these high-quality bi-grams. Specifically, we compute the HR@n to evaluate how often the model's output bi-grams match those in the "true" top n ranked bi-grams given by the final step. Formally, the Hit Ratio@n is given by

$$\text{HR@}n = \frac{1}{n} \sum_{i=1}^{n} \mathbb{I}(\text{the i-th bi-gram output by the current model} \in \text{True\_Top\_n})$$

where $n$ is the number of top predictions being considered and

- $\mathbb{I}$ is the indicator function that returns 1 if the $i$-th bi-gram output by the model is present in the True Top $n$ bi-grams, and 0 otherwise,
- True_Top_n represents the set of "good" bi-grams, which in our case is the set of the top $n$ scoring bi-grams from the final model step.

**Total mass of bi-grams.** In Section I, we use the total mass as a metric to measure the cumulative probabilities of bi-grams from the top 1K bi-grams, weighted by an empirical uni-gram distribution derived from real data. Formally, it is given by: Total Mass $= \sum_{(z_1,z_2)\in\text{Top-1K}} \mathbb{P}_{e_t}(z_2|z_1)\mathbb{P}_{\mathcal{D}}(z_1)$ where:

- $e_t$ is the embedding-unembedding path at the $t$-th pre-training step,
- $(z_1, z_2)$ are the bi-grams being considered,
- $\mathbb{P}_{e_t}(z_2|z_1)$ is the probability assigned by the model $e_t$ (the embedding-unembedding path) for the token $z_2$ given token $z_1$,
- $\mathbb{P}_{\mathcal{D}}(z_1)$ is the probability of $z_1$ under the empirical distribution $\mathcal{D}$, which is the uni-gram probability given by the Infini-gram API (Liu et al., 2024) on the Dolma dataset (Soldaini et al., 2024) (the dataset used to pretrain the model checkpoints).

This metric is designed to evaluate how much "correc" probability mass the model checkpoints assign to bi-grams $(z_1, z_2)$, taking into account the empirical uni-gram probability of $z_1$. It provides insight into how well the model aligns with the empirical distribution of real-world data during the pretraining process.

## G CASE STUDY 1: ANALYZING LLM INTERNALS WITH JET LENS AND JET PATHS (ADDITIONAL RESULTS)

### G.1 ADDITIONAL PLOTS OF JET LENSES

See plots Figures 7 to 17. The details for obtaining the jet lens plots can be found in . Note that for iterative lenses the last block coincides with the model logits for all $k$ by design. We omit the iterative lens for GPT2-large for $k = 2$ due to low cosine similarity.

| | new | _simple | _neural | _architecture | , | _the | _Trans | former |
|---|---|---|---|---|---|---|---|---|
| Block 1 | Supporters | Supporters | Supporters | Supporters | Supporters | Supporters | Supporters | Supporters |
| Block 2 | Supporters | Supporters | Supporters | Supporters | Supporters | Supporters | Supporters | Supporters |
| Block 3 | Supporters | Supporters | Supporters | Supporters | Supporters | Supporters | Supporters | Supporters |
| Block 4 | Supporters | Supporters | Supporters | Supporters | Supporters | Supporters | Supporters | Supporters |
| Block 5 | Supporters | Supporters | Supporters | Supporters | Supporters | Supporters | Supporters | Supporters |
| Block 6 | Supporters | Supporters | Supporters | Supporters | Supporters | Supporters | Supporters | Supporters |
| Block 7 | Supporters | Supporters | Supporters | Supporters | Supporters | Supporters | Supporters | Supporters |
| Block 8 | Supporters | Supporters | Supporters | Supporters | Supporters | Supporters | Supporters | Supporters |
| Block 9 | Supporters | Supporters | Supporters | Supporters | Supporters | Supporters | Supporters | Supporters |
| Block 10 | Supporters | Supporters | Supporters | Supporters | Supporters | Supporters | Supporters | Supporters |
| Block 11 | Supporters | Supporters | Supporters | Supporters | Supporters | Supporters | Supporters | Supporters |
| Block 12 | Supporters | Supporters | Supporters | Supporters | Supporters | Supporters | Supporters | Supporters |
| Block 13 | Supporters | Supporters | Supporters | Supporters | Supporters | Supporters | Supporters | Supporters |
| Block 14 | Supporters | Supporters | Supporters | Supporters | Supporters | Supporters | Supporters | Supporters |
| Block 15 | Supporters | Supporters | Supporters | Supporters | Supporters | Supporters | Supporters | Supporters |
| Block 16 | Supporters | Supporters | Supporters | Supporters | Supporters | Supporters | Supporters | Supporters |
| Block 17 | Supporters | Supporters | Supporters | Supporters | Supporters | Supporters | Supporters | Supporters |
| Block 18 | Supporters | Supporters | Supporters | Supporters | Supporters | Supporters | Supporters | Supporters |
| Block 19 | Supporters | Supporters | Supporters | Supporters | Supporters | Supporters | Supporters | Supporters |
| Block 20 | Supporters | Supporters | Supporters | Supporters | Supporters | Supporters | Supporters | Supporters |
| Block 21 | Supporters | Supporters | Supporters | Supporters | Engineers | Supporters | Supporters | Supporters |
| Block 22 | Supporters | Supporters | Supporters | Supporters | Supporters | Supporters | Supporters | Introduced |
| Block 23 | Supporters | Supporters | Supporters | Supporters | Introduced | Supporters | Supporters | Introduced |
| Block 24 | Supporters | Supporters | Supporters | Supporters | Nonetheless | Nonetheless | Supporters | Introduced |
| Block 25 | Supporters | Supporters | Supporters | Supporters | Attempts | Nonetheless | Supporters | Introduced |
| Block 26 | Supporters | Supporters | Supporters | Supporters | Attempts | Nonetheless | Introduced | Introduced |
| Block 27 | Introduced | Supporters | Supporters | Supporters | Attempts | Nonetheless | Introduced | Introduced |
| Block 28 | Supporters | Supporters | Supporters | Supporters | Attempts | Nonetheless | Introduced | Introduced |
| Block 29 | foreseen | Supporters | Supporters | Supporters | foreseen | Nonetheless | Charges | Introduced |
| Block 30 | foreseen | Supporters | Supporters | Attempts | foreseen | foreseen | Charges | Introduced |
| Block 31 | Supporters | Supporters | Supporters | _for | _the | aminer | former | , |
| Block 32 | _ | _ | _network | _for | _which | _neural | former | , |
| | | | | | | | | |
| Logits | _ | _ | _network | _for | _which | _neural | former | , |

Figure 7: Iterative jet lens ($k = 0$), equivalent to logit lens (nostalgebraist, 2021b), applied over GPT-Neo-2.7B with the input sentence "new simple neural architecture, the Transformer".

| | new | _simple | _neural | _architecture | , | _the | _Trans | former |
|---|---|---|---|---|---|---|---|---|
| Block 1 | , | ton | _network | _for | _which | _first | former | , |
| Block 2 | Supporters | ton | _network | _for | _which | _first | former | , |
| Block 3 | Supporters | ton | _network | _for | _which | _first | former | , |
| Block 4 | Supporters | ton | _network | _for | _which | _first | former | , |
| Block 5 | Supporters | ton | _network | _for | _which | _first | former | , |
| Block 6 | Supporters | ton | _network | _for | _which | _first | former | , |
| Block 7 | Supporters | ton | _network | _for | _which | _first | former | , |
| Block 8 | Supporters | ton | _network | _for | _which | _first | former | , |
| Block 9 | Supporters | ton | _network | _for | _which | _first | former | , |
| Block 10 | Supporters | ton | _network | _for | _which | _first | former | , |
| Block 11 | Supporters | ton | _network | _for | _which | _first | former | , |
| Block 12 | Supporters | ton | _network | _for | _which | _first | former | , |
| Block 13 | Supporters | ton | _network | _for | _which | _first | former | , |
| Block 14 | Supporters | ton | _network | _for | _which | _first | former | , |
| Block 15 | Supporters | ton | _network | _for | _which | _first | former | , |
| Block 16 | Supporters | ton | _network | _for | _which | _first | former | , |
| Block 17 | Supporters | ton | _network | _for | _which | _first | former | , |
| Block 18 | Supporters | ton | _network | _for | _which | _first | former | , |
| Block 19 | Supporters | ton | _network | _for | _which | _first | former | , |
| Block 20 | Supporters | ton | _network | _for | _which | _first | former | , |
| Block 21 | Supporters | ton | _network | _for | _which | _first | former | , |
| Block 22 | Supporters | ton | _network | _for | _which | _first | former | , |
| Block 23 | Supporters | ton | _network | _for | _which | _first | former | , |
| Block 24 | Supporters | ton | _network | _for | _which | _so | former | , |
| Block 25 | Supporters | ton | _network | _for | _which | _first | former | , |
| Block 26 | Supporters | ton | _network | _for | _which | _first | former | , |
| Block 27 | Supporters | ton | _network | _for | _which | _first | former | , |
| Block 28 | Supporters | ton | _network | _for | _which | _first | former | , |
| Block 29 | foreseen | ton | _network | _for | _which | _first | former | , |
| Block 30 | foreseen | ton | _network | _for | _which | _first | former | , |
| Block 31 | Supporters | _ | _network | _for | _which | _first | former | , |
| Block 32 | _ | _ | _network | _for | _which | _neural | former | , |
| | | | | | | | | |
| Logits | _ | _ | _network | _for | _which | _neural | former | , |

Figure 8: Iterative jet lens ($k = 1$), applied over GPT-Neo-2.7B with the input sentence "new simple neural architecture, the Transformer".

|  | new | _simple | _neural | _architecture | , | _the | _Trans | former |
|---|---|---|---|---|---|---|---|---|
| Block 1 | _the | _ | _nets | !: | _âĢ¦" | _âĢ¦" | former | !: |
| Block 2 | _the | _ | _network | _outper | âĢ¦" | _âĢ¦" | former | _[ |
| Block 3 | _the | _ | _network | _for | _trained | _Conv | former | _[ |
| Block 4 | _the | _ | _network | _for | _the | _Conv | former | , |
| Block 5 | _the | _ | _network | _for | _the | _neural | former | , |
| Block 6 | _the | _ | _network | _for | _the | _neural | former | , |
| Block 7 | _the | _ | _network | _for | _the | _architecture | former | , |
| Block 8 | _the | _ | _network | _for | _the | _architecture | former | , |
| Block 9 | _the | _ | _network | _for | _the | _architecture | former | , |
| Block 10 | _the | _ | _network | _for | _the | _architecture | former | , |
| Block 11 | _the | _ | _network | _for | _the | _architecture | former | , |
| Block 12 | _the | _ | _network | _for | _the | _architecture | former | , |
| Block 13 | _the | _ | _network | _for | _the | _architecture | former | , |
| Block 14 | _the | _ | _network | _for | _the | _neural | former | , |
| Block 15 | _the | _ | _network | _for | _the | _neural | former | , |
| Block 16 | _the | _ | _network | _for | _the | _neural | former | , |
| Block 17 | _the | _ | _network | _for | _the | _neural | former | , |
| Block 18 | _the | _ | _network | _for | _the | _neural | former | , |
| Block 19 | _the | _ | _network | _for | _the | _neural | former | , |
| Block 20 | _the | _ | _network | _for | _the | _neural | former | , |
| Block 21 | _the | _ | _network | _for | _the | _neural | former | , |
| Block 22 | _the | _ | _network | _for | _the | _neural | former | , |
| Block 23 | _the | _ | _network | _for | _the | _neural | former | , |
| Block 24 | _the | _ | _network | _for | _the | _neural | former | , |
| Block 25 | _the | _ | _network | _for | _the | _neural | former | , |
| Block 26 | _the | _ | _network | _for | _the | _neural | former | , |
| Block 27 | _the | _ | _network | _for | _the | _neural | former | , |
| Block 28 | _the | _ | _network | _for | _the | _neural | former | , |
| Block 29 | _the | _ | _network | _for | _the | _neural | former | , |
| Block 30 | _the | _ | _network | _for | _and | _neural | former | , |
| Block 31 | , | _ | _network | _for | _and | _neural | former | , |
| Block 32 | _ | _ | _network | _for | _which | _neural | former | , |
| Logits | _ | _ | _network | _for | _which | _neural | former | , |

Figure 9: Iterative jet lens ($k = 2$), applied over GPT-Neo-2.7B with the input sentence "new simple neural architecture, the Transformer"

|  | new | _simple | _neural | _architecture | , | _the | _Trans | former |
|---|---|---|---|---|---|---|---|---|
| Block 1 | bie | _simple | _neural | _architecture | _and | _the | fig | former |
| Block 2 | bie | _simple | _neural | _architecture | _and | _main | ient | former |
| Block 3 | bie | _simple | _neural | _architecture | _and | _new | ient | former |
| Block 4 | bie | _way | _neural | _architecture | _and | _first | ient | _titan |
| Block 5 | bie | _way | _networks | _architecture | _and | _next | ient | _Prime |
| Block 6 | bie | _networks | _networks | _architecture | _and | _next | ient | _Matrix |
| Block 7 | _href | _enough | _networks | _architecture | _and | _first | ient | _Prime |
| Block 8 | _iTunes | _enough | _neural | _architecture | _which | _first | ient | _Revolution |
| Block 9 | , | _enough | _neural | _architecture | _which | _first | ient | _Prime |
| Block 10 | , | _enough | _network | _architecture | _which | _first | ient | _Revolution |
| Block 11 | , | _enough | _network | _model | _which | _only | ient | _Pro |
| Block 12 | , | _enough | _network | _architecture | _which | _only | ient | _Pro |
| Block 13 | , | _enough | _network | _model | _which | _first | ient | _Pro |
| Block 14 | , | _enough | _network | _model | _which | _first | ient | _Pro |
| Block 15 | , | _enough | _network | _model | _which | _only | ient | _Pro |
| Block 16 | , | - | _network | _model | _which | _only | ient | _Revolution |
| Block 17 | , | - | _system | _model | _which | _only | ient | _Prime |
| Block 18 | , | - | _system | _model | _which | _only | ient | _Prime |
| Block 19 | , | - | _system | _model | _which | _only | ient | _Prime |
| Block 20 | , | - | _system | _model | _which | _only | ient | _Prime |
| Block 21 | , | - | _system | _model | _which | _only | ient | _Prime |
| Block 22 | , | - | _network | _model | _which | _only | ient | _Prime |
| Block 23 | , | ton | _network | _model | _which | _only | ient | _Prime |
| Block 24 | , | ton | _network | _model | _which | _only | ient | _Prime |
| Block 25 | , | ton | _network | _model | _which | _first | ient | _Prime |
| Block 26 | , | ton | _network | _model | _which | _only | ient | _Prime |
| Block 27 | , | ton | _network | _for | _which | _first | ient | _Prime |
| Block 28 | , | - | _network | " | _which | _only | ient | _Prime |
| Block 29 | , | - | _network | " | _which | _neural | ient | _Prime |
| Block 30 | , | " | _network | " | _which | _neural | ient | , |
| Block 31 | , | " | _network | " | _which | _neural | ient | , |
| Block 32 | , | " | _network | " | _which | _neural | ient | , |
| Block 33 | , | " | _network | _for | _which | _neural | ient | , |
| Block 34 | , | " | _network | ' | _which | _neural | ient | , |
| Block 35 | , | " | _network | ' | _which | _neural | c | , |
| Block 36 | _ | " | _network | ' | _which | _neural | c | , |
| Logits | _ | " | _network | ' | _which | _neural | c | , |

Figure 10: Iterative jet lens ($k = 0$), equivalent to Logit Lens (nostalgebraist, 2021b), applied over GPT-2-large with the input sentence "new simple neural architecture, the Transformer".

|  | new | _simple | _neural | _architecture | , | _the | _Trans | former |
|---|---|---|---|---|---|---|---|---|
| Block 1 | bie | " | _network | " | _which | _neural | c | _is |
| Block 2 | bie | " | _network | ' | _which | _neural | c | _is |
| Block 3 | bie | " | _network | ' | _which | _neural | c | _is |
| Block 4 | _ | " | _network | ' | _which | _neural | c | _is |
| Block 5 | _ | " | _network | ' | _which | _neural | c | _is |
| Block 6 | _ | " | _network | ' | _which | _neural | c | _is |
| Block 7 | _ | " | _network | ' | _which | _neural | c | _is |
| Block 8 | _ | " | _network | ' | _which | _neural | c | _is |
| Block 9 | _ | " | _network | ' | _which | _neural | c | _is |
| Block 10 | , | " | _network | ' | _which | _neural | c | _is |
| Block 11 | , | " | _network | ' | _which | _neural | c | _is |
| Block 12 | , | " | _network | ' | _which | _neural | c | , |
| Block 13 | , | " | _network | ' | _where | _neural | c | , |
| Block 14 | , | " | _network | ' | _and | _neural | c | , |
| Block 15 | , | " | _network | ' | _and | _neural | c | , |
| Block 16 | , | " | _network | ' | _and | _neural | c | , |
| Block 17 | , | " | _network | ' | _and | _neural | c | , |
| Block 18 | , | " | _network | ' | _and | _neural | c | , |
| Block 19 | , | " | _network | ' | _and | _neural | c | , |
| Block 20 | , | " | _network | ' | _and | _neural | c | , |
| Block 21 | , | " | _network | ' | _and | _neural | c | , |
| Block 22 | , | " | _network | ' | _and | _neural | c | , |
| Block 23 | , | " | _network | ' | _the | _neural | c | , |
| Block 24 | , | " | _network | ' | _and | _neural | c | , |
| Block 25 | , | " | _network | ' | _and | _neural | c | , |
| Block 26 | , | " | _network | ' | _and | _neural | c | , |
| Block 27 | , | " | _network | ' | _and | _neural | c | , |
| Block 28 | , | " | _network | ' | _and | _neural | c | , |
| Block 29 | , | " | _network | ' | _and | _human | c | , |
| Block 30 | , | " | _network | ' | _and | _same | c | , |
| Block 31 | , | " | _network | ' | _and | _same | c | , |
| Block 32 | , | " | _network | ' | _and | _same | c | , |
| Block 33 | , | " | _network | ' | _and | _neural | c | , |
| Block 34 | , | " | _network | ' | _which | _neural | c | , |
| Block 35 | - | " | _network | ' | _which | _neural | c | , |
| Block 36 | _ | " | _network | ' | _which | _neural | c | , |
| Logits | _ | " | _network | ' | _which | _neural | c | , |

Figure 11: Iterative jet lens ($k = 1$), applied over GPT-2-large with the input sentence "new simple neural architecture, the Transformer"

|  | new | _simple | _neural | _architecture | , | _the | _Trans | former |
|---|---|---|---|---|---|---|---|---|
| Block 1 (4.40%) | , (6.62%) | _simple (3.91%) | _neural (4.42%) | _architecture (3.97%) | _which (4.07%) | _same (4.37%) | cend (3.93%) | former (3.91%) |
| Block 2 (4.15%) | , (6.59%) | _retro (3.85%) | _prog (4.32%) | _error (3.74%) | _including (3.93%) | _resulting (4.14%) | ference (3.69%) | _Robo (2.99%) |
| Block 3 (4.23%) | , (6.59%) | ove (4.13%) | _Matter (4.12%) | killer (3.51%) | _which (4.00%) | _AVG (4.01%) | em (3.56%) | Mars (3.91%) |
| Block 4 (4.11%) | _the (6.59%) | _reg (3.51%) | lect (4.37%) | OX (3.68%) | _found (4.05%) | netflix (4.09%) | Charge (2.95%) | Â® (3.69%) |
| Block 5 (6.11%) | , (6.59%) | ware (3.54%) | _product (3.68%) | _towards (3.70%) | _evolution (3.88%) | _ones (3.74%) | it (20.20%) | _Mant (3.57%) |
| Block 6 (3.91%) | , (6.58%) | ies (3.59%) | _networks (4.11%) | _developed (3.45%) | _developed (3.55%) | _Mehran (3.45%) | ition (3.54%) | bur (3.01%) |
| Block 7 (4.00%) | , (6.56%) | face (3.75%) | _studies (3.88%) | _based (3.52%) | _hackers (3.76%) | _Turing (3.73%) | _Series (2.97%) | _Suite (3.83%) |
| Block 8 (4.06%) | , (6.42%) | key (3.83%) | _model (4.18%) | _based (3.53%) | _requiring (3.49%) | _algorithm (4.14%) | ient (3.62%) | _II (3.25%) |
| Block 9 (4.09%) | , (7.45%) | _clutter (4.08%) | _model (3.69%) | _test (3.40%) | _which (3.11%) | _neural (3.55%) | verse (3.82%) | _Cube (3.66%) |
| Block 10 (10.50%) | . (16.50%) | lists (9.61%) | g (4.99%) | _of (16.60%) | _which (11.47%) | _neural (5.79%) | _neural (3.50%) | _is (15.56%) |
| Block 11 (25.30%) | , (16.96%) | " (27.59%) | _networks (28.89%) | " (24.52%) | _the (26.92%) | _new (29.14%) | m (22.95%) | _neural (25.40%) |
| Block 12 (25.13%) | , (6.56%) | . (28.62%) | net (29.35%) | , (26.40%) | the (27.77%) | the (29.85%) | c (25.27%) | . (27.23%) |
| Logits | , | - | _network | _that | _which | _neural | ient | _is |
| Expan. (1.000) | , | - | _network | _of | _which | " | - | _is |

Figure 12: Joint jet lens with learnable weightings ($k = 0$), applied over GPT2 with the input sentence "new simple neural architecture, the Transformer"

|  | new | _simple | _neural | _architecture | , | _the | _Trans | former |
|---|---|---|---|---|---|---|---|---|
| Block 1 (15.30%) | . (7.49%) | " (16.78%) | _networks (16.96%) | ", (18.37%) | _neural (14.61%) | _neural (14.05%) | verse (16.45%) | _Neural (17.73%) |
| Block 2 (4.57%) | , (13.81%) | json (3.21%) | _networks (3.29%) | _model (3.46%) | _which (3.11%) | _neural (3.02%) | cend (3.23%) | _Neural (3.45%) |
| Block 3 (4.49%) | , (14.25%) | tons (3.25%) | _networks (2.82%) | _architecture (3.32%) | _neural (3.10%) | _neural (3.00%) | porter (3.03%) | _Neural (3.17%) |
| Block 4 (4.10%) | . (11.55%) | tons (3.28%) | _networks (3.27%) | _leveraging (3.19%) | _synt (3.04%) | _neural (2.98%) | verse (2.90%) | _Neural (2.57%) |
| Block 5 (4.02%) | . (9.58%) | tons (3.05%) | _networks (3.25%) | _algorithm (3.45%) | _which (3.14%) | _neural (2.99%) | mitter (3.24%) | _Neural (3.47%) |
| Block 6 (3.02%) | . (2.75%) | _linkage (2.65%) | _net (3.04%) | _algorithms (3.26%) | _detecting (2.94%) | _neural (2.80%) | cend (3.30%) | _Neural (3.45%) |
| Block 7 (2.91%) | . (2.98%) | _teleportation (2.78%) | _nets (3.19%) | _approach (3.24%) | _specifically (2.49%) | _cortex (2.58%) | genic (3.07%) | _Cortex (2.95%) |
| Block 8 (4.60%) | bid (3.10%) | nex (7.64%) | _network (2.63%) | _platform (2.62%) | _neural (4.81%) | _participant (9.06%) | cription (3.50%) | _Neural (3.45%) |
| Block 9 (7.44%) | iaries (3.10%) | url (5.60%) | _networks (7.77%) | _intelligence (4.86%) | _Torch (14.64%) | _welcoming (13.48%) | Secure (7.21%) | _conv (2.83%) |
| Block 10 (15.04%) | akings (13.99%) | widget (14.80%) | _network (16.20%) | _None (13.05%) | _Bund (15.37%) | _safest (14.72%) | cend (16.11%) | _disabling (16.06%) |
| Block 11 (16.50%) | ity (3.19%) | ton (18.47%) | _network (18.79%) | _architecture (20.49%) | _which (16.34%) | _neural (15.62%) | istor (18.84%) | â†¢ (20.28%) |
| Block 12 (18.00%) | , (14.21%) | - (18.49%) | network (18.78%) | that (20.68%) | which (16.41%) | neural (15.70%) | ient (19.11%) | is (20.60%) |
| Logits | , | - | _network | _that | _which | _neural | ient | _is |
| Expan. (1.000) | akings | json | _networks | _framework | _neural | _neural | cend | _Neural |

Figure 13: Joint jet lens with learnable weightings ($k = 1$), applied over GPT2 with the input sentence "new simple neural architecture, the Transformer"

| | new | _simple | _neural | _architecture | , | _the | _Trans | former |
|---|---|---|---|---|---|---|---|---|
| Block 1 (3.58%) | Supporters (1.55%) | Supporters (3.24%) | Supporters (3.46%) | Supporters (5.37%) | Supporters (5.08%) | Supporters (3.52%) | Supporters (3.88%) | Supporters (2.56%) |
| Block 2 (2.13%) | foreseen (1.61%) | foreseen (2.97%) | foreseen (1.15%) | Introduced (3.96%) | foreseen (1.09%) | foreseen (1.54%) | Supporters (3.67%) | Supporters (1.03%) |
| Block 3 (2.07%) | Amid (1.65%) | Supporters (2.01%) | Across (1.32%) | gewater (1.14%) | Supporters (3.66%) | Supporters (2.93%) | Supporters (2.58%) | leground (1.28%) |
| Block 4 (1.57%) | _impover (1.97%) | _unpop (2.18%) | _unpop (1.46%) | _impover (1.33%) | _impover (1.39%) | _impover (1.71%) | _uphe (1.27%) | _impover (1.27%) |
| Block 5 (1.47%) | Attempts (1.76%) | _municip (2.15%) | _airst (1.45%) | _linem (1.29%) | amiliar (1.32%) | pelling (1.38%) | rieving (1.26%) | _linem (1.13%) |
| Block 6 (1.45%) | Residents (1.76%) | _athlet (2.17%) | rha (1.44%) | _twent (1.34%) | _way (1.05%) | ters (1.40%) | rha (1.23%) | _Xuan (1.25%) |
| Block 7 (3.57%) | Ironically (1.63%) | celona (2.74%) | wrap (3.78%) | _look (5.71%) | _airstrike (1.22%) | _equivalent (2.63%) | _different (6.30%) | _hollow (4.58%) |
| Block 8 (4.63%) | Supporters (1.61%) | imura (3.91%) | vantage (3.03%) | anoia (5.48%) | foreseen (6.13%) | ileen (4.55%) | Enlarge (5.70%) | assador (6.59%) |
| Block 9 (3.14%) | Ironically (1.65%) | erguson (2.00%) | certain (2.53%) | OUR (1.28%) | _local (3.54%) | erguson (1.80%) | enter (5.43%) | bec (6.89%) |
| Block 10 (1.73%) | foreseen (1.65%) | foreseen (2.01%) | Engineers (1.20%) | Engineers (2.88%) | asury (1.19%) | thinkable (1.40%) | Attempts (2.53%) | uddenly (0.96%) |
| Block 11 (1.71%) | likely (1.57%) | extremely (1.88%) | aples (1.18%) | _screenplay (1.29%) | earances (1.30%) | earances (4.13%) | oother (1.20%) | _resurg (1.12%) |
| Block 12 (4.53%) | Ironically (1.73%) | Phones (3.91%) | ADVERTISEMENT (4.39%) | ADVERTISEMENT (6.03%) | isively (4.65%) | _Blvd (4.46%) | ADVERTISEMENT (6.08%) | ADVERTISEMENT (4.99%) |
| Block 13 (2.80%) | _a (1.68%) | aji (2.83%) | imbabwe (1.33%) | rone (1.28%) | OTOS (5.38%) | ppard (3.08%) | ppard (1.07%) | aji (5.76%) |
| Block 14 (2.91%) | foreseen (1.66%) | ADVERTISEMENT (1.83%) | Marginal (3.82%) | chell (1.32%) | _Appalach (1.33%) | _Caucasus (4.66%) | _still (5.47%) | , (3.23%) |
| Block 15 (1.47%) | ormons (1.78%) | _confir (1.89%) | uring (1.34%) | ured (1.25%) | _AoE (1.38%) | _Caucas (1.43%) | _lineman (1.25%) | _topple (1.22%) |
| Block 16 (3.98%) | Against (1.82%) | folios (1.93%) | @ (6.49%) | thinkable (3.49%) | _tsun (1.26%) | _D (4.65%) | I (5.84%) | arsh (6.38%) |
| Block 17 (2.89%) | urses (1.38%) | untied (4.46%) | ortunate (3.72%) | ithub (1.21%) | _our (4.69%) | ortment (1.51%) | erenn (4.91%) | ombies (1.21%) |
| Block 18 (5.12%) | foreseen (1.63%) | Supporters (4.53%) | Nonetheless (6.62%) | Ironically (5.07%) | Thankfully (5.66%) | Shortly (4.52%) | af (5.80%) | _is (7.12%) |
| Block 19 (2.96%) | pherd (1.47%) | _enough (4.91%) | ag (3.58%) | _for (5.69%) | incerity (1.08%) | incerity (2.75%) | extreme (3.01%) | phabet (1.21%) |
| Block 20 (5.68%) | Ć (2.06%) | Ć (5.07%) | _just (7.05%) | Ć (6.91%) | Attempts (6.51%) | paralleled (4.49%) | - (6.53%) | , (6.87%) |
| Block 21 (1.46%) | ription (1.60%) | ription (2.15%) | _Playoffs (1.48%) | isdom (1.06%) | _frontrunner (1.36%) | _frontrunner (1.69%) | _TBD (1.24%) | pered (1.06%) |
| Block 22 (4.55%) | _in (3.36%) | _first (5.29%) | _two (7.06%) | _one (6.98%) | _which (6.97%) | _one (4.56%) | isEnabled (1.03%) | elligence (1.15%) |
| Block 23 (5.21%) | , (4.80%) | 1] (5.23%) | _* (7.13%) | ) (6.26%) | _while (6.31%) | _point (4.57%) | albeit (1.15%) | B (6.21%) |
| Block 24 (6.13%) | _a (5.62%) | _m (5.26%) | _first (7.18%) | _for (7.33%) | _for (7.33%) | _so (4.70%) | _trans (5.70%) | rieving (5.90%) |
| Block 25 (1.55%) | foreseen (1.67%) | acly (2.14%) | _enthus (1.49%) | _anecd (1.35%) | _trainers (1.43%) | _subreddits (1.74%) | ithub (1.28%) | _Trainer (1.27%) |
| Block 26 (2.61%) | - (6.25%) | _simple (2.08%) | _simple (5.95%) | ername (1.30%) | haar (1.34%) | _satell (1.74%) | igsaw (1.02%) | _headphone (1.17%) |
| Block 27 (2.65%) | âĞ (7.40%) | âĞ (5.48%) | _DSM (1.35%) | held (1.30%) | dayName (1.38%) | _artif (1.75%) | --+ (1.27%) | _nostalg (1.30%) |
| Block 28 (2.39%) | _fps (8.56%) | >>\ (2.30%) | _Oo (1.42%) | _tacos (1.30%) | _msec (1.41%) | _unbeliev (1.75%) | _hrs (1.12%) | _reminis (1.28%) |
| Block 29 (1.97%) | âĞ¦* (5.17%) | _convol (2.18%) | ricanes (1.47%) | _Gujar (1.25%) | acerb (1.38%) | cffff (1.74%) | _negoti (1.28%) | _automakers (1.27%) |
| Block 30 (1.84%) | âĞ¦* (4.01%) | _anecd (2.24%) | _unve (1.49%) | _overwhel (1.37%) | !?* (1.43%) | 20439 (1.78%) | _negoti (1.29%) | _calculates (1.12%) |
| Block 31 (4.61%) | !!* (8.40%) | âĞ¦* (2.57%) | _greets (1.35%) | _entert (1.80%) | \\\\ (4.44%) | \\\\ (6.14%) | *! (5.27%) | '/ (6.88%) |
| Block 32 (5.64%) | âĞ¦* (9.55%) | !?* (4.42%) | âĞ¦* (2.29%) | âĞ¦* (5.37%) | âĞ¦* (6.35%) | _\' (9.03%) | ©¶æ¥µ (3.34%) | âĞ¦* (4.75%) |
| Logits | _ | _ | _network | _for | _which | _neural | former | , |
| Expan. (0.977) | _the | _and | - | _for | _the | _first | - | , |

Figure 14: Joint jet lens with learnable weightings ($k = 0$), applied over GPT-Neo-2.7B with the input sentence "new simple neural architecture, the Transformer"

| | new | _simple | _neural | _architecture | , | _the | _Trans | former |
|---|---|---|---|---|---|---|---|---|
| Block 1 (7.36%) | , (3.40%) | ton (8.06%) | _network (8.57%) | _for (8.22%) | _which (7.51%) | _first (7.30%) | former (7.43%) | , (8.36%) |
| Block 2 (4.83%) | - (2.39%) | _ (5.23%) | _network (6.91%) | _for (4.98%) | _which (4.60%) | _neural (4.77%) | former (5.09%) | , (4.68%) |
| Block 3 (1.31%) | _File (1.62%) | _ (1.29%) | _network (1.31%) | | _which (1.25%) | _CNN (1.22%) | former (1.20%) | , (1.32%) |
| Block 4 (7.81%) | _impover (5.74%) | _unpop (8.48%) | _impover (8.76%) | _impover (8.45%) | _impover (7.67%) | _Neural (7.51%) | former (7.39%) | _Networks (8.45%) |
| Block 5 (1.79%) | User (5.29%) | _ (1.31%) | _network (1.30%) | _for (1.29%) | _which (1.29%) | _neural (1.26%) | former (1.25%) | , (1.31%) |
| Block 6 (1.79%) | Instance (5.33%) | _ (1.33%) | _network (1.31%) | _for (1.29%) | _which (1.26%) | _neural (1.23%) | former (1.23%) | , (1.32%) |
| Block 7 (1.59%) | File (3.56%) | _ (1.37%) | _network (1.36%) | _for (1.33%) | _which (1.28%) | _neural (1.24%) | former (1.25%) | , (1.32%) |
| Block 8 (1.70%) | Supporters (5.02%) | _ (1.29%) | _network (1.28%) | _for (1.25%) | _which (1.24%) | _Neural (1.17%) | former (1.12%) | , (1.21%) |
| Block 9 (1.77%) | Enlarge (5.04%) | _ (1.37%) | _network (1.37%) | _for (1.32%) | _which (1.26%) | _neural (1.23%) | former (1.25%) | , (1.31%) |
| Block 10 (4.41%) | foreseen (5.36%) | _ (5.77%) | _network (6.19%) | _for (5.99%) | _which (1.15%) | _neural (0.93%) | former (2.45%) | , (7.42%) |
| Block 11 (1.31%) | , (1.90%) | _ (1.30%) | _network (1.29%) | _for (1.20%) | _which (1.18%) | _neural (1.19%) | former (1.19%) | , (1.24%) |
| Block 12 (1.21%) | , (1.74%) | _ (1.11%) | _network (1.17%) | _for (1.10%) | _which (1.16%) | _neural (1.14%) | former (1.07%) | , (1.21%) |
| Block 13 (1.37%) | _ (1.94%) | _ (1.36%) | _network (1.35%) | _for (1.32%) | _which (1.23%) | _neural (1.21%) | former (1.23%) | , (1.32%) |
| Block 14 (1.22%) | , (1.82%) | _ (1.18%) | _network (1.22%) | _for (1.24%) | _which (1.15%) | _neural (1.09%) | former (1.04%) | , (1.12%) |
| Block 15 (1.34%) | _ (1.90%) | _ (1.33%) | _network (1.31%) | _for (1.29%) | _which (1.21%) | _neural (1.20%) | former (1.20%) | , (1.28%) |
| Block 16 (1.31%) | ( (1.91%) | _ (1.28%) | _network (1.28%) | _for (1.24%) | _which (1.18%) | _neural (1.19%) | former (1.18%) | _model (1.23%) |
| Block 17 (1.31%) | _ (1.90%) | _ (1.29%) | _network (1.28%) | _for (1.26%) | _which (1.14%) | _neural (1.12%) | former (1.16%) | , (1.29%) |
| Block 18 (4.55%) | , (1.65%) | _ (5.16%) | _network (3.55%) | _for (5.49%) | _which (6.28%) | _neural (6.05%) | former (5.05%) | , (3.17%) |
| Block 19 (1.24%) | , (1.84%) | _ (1.23%) | _network (1.17%) | _for (1.18%) | _which (1.23%) | _neural (0.97%) | former (1.10%) | _model (1.18%) |
| Block 20 (3.30%) | Ć (1.84%) | _ (2.30%) | _network (1.16%) | _for (4.21%) | _which (6.29%) | _neural (5.89%) | former (2.70%) | _architecture (2.00%) |
| Block 21 (1.87%) | _ (1.80%) | _ (1.21%) | _network (1.12%) | _for (1.15%) | _which (3.82%) | _neural (3.71%) | former (1.10%) | , (1.02%) |
| Block 22 (4.81%) | - (1.91%) | _infographic (8.14%) | _network (3.50%) | _outper (3.92%) | _which (6.89%) | _neural (6.76%) | former (1.57%) | _[ (3.83%) |
| Block 23 (2.01%) | , (1.91%) | _ (1.14%) | _network (1.40%) | _learns (1.38%) | _which (3.94%) | _Conv (3.99%) | former (1.14%) | _model (1.18%) |
| Block 24 (6.02%) | , (1.94%) | _infographic (8.04%) | _network (7.20%) | _unve (8.00%) | _unve (7.47%) | _Neural (7.02%) | former (3.53%) | _model (4.98%) |
| Block 25 (1.19%) | _ (1.87%) | _ (1.19%) | _network (1.09%) | _which (0.96%) | _which (1.25%) | _âĞ (1.07%) | former (1.06%) | , (1.04%) |
| Block 26 (1.55%) | _ (1.89%) | _ (1.18%) | _network (2.18%) | _called (1.22%) | _which (1.25%) | _Conv (1.09%) | former (2.57%) | , (1.06%) |
| Block 27 (2.23%) | _ (1.93%) | ton (3.53%) | _network (1.09%) | _for (1.21%) | _which (0.99%) | _model (1.13%) | former (6.67%) | , (1.25%) |
| Block 28 (2.76%) | _ (1.73%) | json (1.02%) | _network (3.49%) | _for (1.84%) | _which (0.95%) | _Neural (3.31%) | former (6.31%) | , (3.42%) |
| Block 29 (3.22%) | âĞ¦* (6.01%) | _ (1.32%) | _network (1.00%) | _for (1.01%) | _and (1.74%) | _neural (1.90%) | oother (7.25%) | , (5.54%) |
| Block 30 (6.24%) | âĞ¦* (6.04%) | _ (3.56%) | _network (7.34%) | _for (5.45%) | _which (6.05%) | _neural (6.14%) | former (7.30%) | ÄI (8.04%) |
| Block 31 (7.76%) | !!* (5.96%) | _ (8.27%) | _network (8.68%) | _for (8.36%) | _the (7.67%) | _Conv (7.46%) | former (7.35%) | , (8.37%) |
| Block 32 (7.84%) | âĞ¦* (5.81%) | !?* (8.35%) | _network (8.78%) | , (8.43%) | _and (7.70%) | _neural (7.51%) | former (7.57%) | _model (8.53%) |
| Logits | _ | _ | _network | _for | _which | _neural | former | , |
| Expan. (0.993) | _ | _ | _network | _for | _which | _neural | former | , |

Figure 15: Joint jet lens with learnable weightings ($k = 1$), applied over GPT-Neo-2.7B with the input sentence "new simple neural architecture, the Transformer"

| | new | _simple | _neural | _architecture | , | _the | _Trans | former |
|---|---|---|---|---|---|---|---|---|
| Block 1 (3.19%) | bie (4.48%) | _simple (4.99%) | _neural (0.98%) | _architecture (1.08%) | _and (5.08%) | _the (5.85%) | fig (2.07%) | former (1.01%) |
| Block 2 (1.81%) | _arrivals (2.43%) | tons (1.22%) | _rack (3.83%) | _model (1.07%) | _the (1.01%) | _main (1.01%) | ient (3.10%) | _generation (0.85%) |
| Block 3 (2.49%) | _entry (5.53%) | _fitting (5.41%) | _clusters (3.05%) | _det (1.14%) | _thanks (0.99%) | _second (1.00%) | cription (0.97%) | _barrier (1.86%) |
| Block 4 (3.02%) | bies (3.47%) | _private (5.64%) | _env (5.41%) | _clusters (1.18%) | _aspirin (1.09%) | _hypothesis (1.08%) | cript (5.55%) | _Mund (0.75%) |
| Block 5 (1.75%) | _mansion (3.47%) | _Transcript (1.03%) | ous (2.48%) | _suit (1.15%) | chuk (1.11%) | _Oracle (1.17%) | _Card (2.55%) | cknow (1.00%) |
| Block 6 (1.84%) | _Parables (2.46%) | _Bald (1.45%) | lzer (0.99%) | sche (1.21%) | %); (1.11%) | ija (1.18%) | ione (5.34%) | atti (1.01%) |
| Block 7 (2.51%) | DERR (2.47%) | _sp (1.62%) | _wired (3.21%) | _experiments (0.89%) | )* (1.02%) | _gloss (1.17%) | aways (4.96%) | _system (4.48%) |
| Block 8 (1.80%) | , (2.32%) | _Tall (1.04%) | _experiments (0.89%) | MIT (1.21%) | mac (1.06%) | fts (1.16%) | rock (5.75%) | con (0.97%) |
| Block 9 (1.79%) | , (2.19%) | onel (1.11%) | _layer (5.70%) | _hum (1.10%) | arily (1.06%) | _Hots (1.20%) | iter (0.98%) | _boxes (0.96%) |
| Block 10 (2.17%) | , (2.18%) | tested (1.09%) | / (6.21%) | _deployed (1.18%) | _disrupt (3.01%) | ew (1.11%) | _INS (0.76%) | _Drive (1.80%) |
| Block 11 (1.20%) | , (2.18%) | azon (1.10%) | âh³âH_ (1.00%) | ea (1.20%) | Ro (1.10%) | _Dive (1.10%) | _Revised (0.95%) | _Prol (1.00%) |
| Block 12 (1.17%) | , (2.20%) | _Think (1.05%) | _Dish (0.86%) | _Layer (1.11%) | _Sing (0.99%) | uts (1.16%) | _button (0.94%) | _proble (1.02%) |
| Block 13 (1.88%) | _and (2.22%) | _ab (2.77%) | ourt (4.71%) | _Malf (1.20%) | _REPL (0.99%) | _naked (1.17%) | oran (0.98%) | _cred (1.01%) |
| Block 14 (1.60%) | _and (2.22%) | alg (1.06%) | _underestimated (0.97%) | _percentile (1.19%) | _which (2.35%) | _nonetheless (1.15%) | igo (3.05%) | _Hut (0.81%) |
| Block 15 (2.19%) | _and (2.24%) | - (4.45%) | _Subst (1.01%) | chan (1.16%) | ATURES (1.09%) | _hitch (1.19%) | _Mini (0.99%) | _Bre (5.41%) |
| Block 16 (2.24%) | _and (2.26%) | _image (5.83%) | _cell (4.89%) | _packs (1.05%) | _marked (0.91%) | _Finn (1.09%) | omes (0.89%) | _Cipher (0.99%) |
| Block 17 (1.72%) | _and (2.27%) | _Â± (1.11%) | _formulation (0.96%) | isen (1.22%) | _modular (1.08%) | _Space (0.99%) | _Neural (0.85%) | _Trainer (5.29%) |
| Block 18 (1.54%) | _and (2.21%) | _bond (1.06%) | _IPM (1.01%) | _{ (4.36%) | build (0.97%) | plex (1.04%) | brand (0.78%) | _Quest (0.91%) |
| Block 19 (2.17%) | _and (2.13%) | _cross (3.75%) | _proceeds (5.61%) | _named (2.11%) | _called (0.93%) | parallel (1.08%) | Shares (0.96%) | _lost (0.81%) |
| Block 20 (2.64%) | , (3.62%) | ": (0.98%) | rons (1.15%) | _Neural (2.26%) | _coupled (4.39%) | _omm (2.30%) | fect (4.73%) | _Fly (1.73%) |
| Block 21 (1.27%) | , (3.47%) | _ft (0.97%) | ysis (1.03%) | _template (1.09%) | _with (0.83%) | _latter (1.09%) | adic (0.79%) | âHç (0.87%) |
| Block 22 (3.88%) | , (3.56%) | types (0.98%) | _Turing (2.15%) | . (7.00%) | _which (4.55%) | _most (5.96%) | gress (1.06%) | _VT (5.74%) |
| Block 23 (3.17%) | , (3.95%) | tv (1.07%) | blade (0.96%) | _...* (1.16%) | _i (2.87%) | _model (5.98%) | du (4.83%) | _erg (4.52%) |
| Block 24 (5.36%) | , (3.89%) | _prayers (5.37%) | _Turing (6.05%) | , (6.95%) | _which (5.59%) | _brain (6.37%) | Memory (5.62%) | als (3.00%) |
| Block 25 (2.84%) | , (3.80%) | _complex (0.86%) | _surgery (0.93%) | * (0.97%) | _Neural (1.57%) | _one (5.52%) | _EEG (3.47%) | , (5.60%) |
| Block 26 (5.61%) | , (3.63%) | _dot (6.73%) | _Turing (6.16%) | _for (7.62%) | _then (6.26%) | _Neural (5.36%) | ocy (5.16%) | _robot (3.94%) |
| Block 27 (4.91%) | , (3.64%) | ?" (7.12%) | _algorithm (2.21%) | ", (6.61%) | _where (5.86%) | _so (5.87%) | vier (1.80%) | _or (6.21%) |
| Block 28 (3.91%) | , (2.94%) | _solution (0.91%) | _simulation (4.19%) | *, (5.57%) | _which (5.77%) | _F (6.14%) | imil (0.95%) | _Mega (4.63%) |
| Block 29 (4.07%) | , (1.51%) | _life (6.69%) | _network (2.58%) | ] (2.36%) | _using (5.32%) | _neural (6.09%) | Washington (4.30%) | _brains (3.73%) |
| Block 30 (5.05%) | , (1.96%) | ÂL (5.52%) | _net (5.50%) | _that (7.83%) | _neural (6.24%) | _neural (6.05%) | _underground (4.91%) | _Brain (2.39%) |
| Block 31 (5.02%) | , (2.04%) | " (6.84%) | _Machine (3.46%) | _* (7.99%) | _neural (6.56%) | _neural (6.10%) | onet (0.95%) | _neural (6.19%) |
| Block 32 (5.00%) | , (2.06%) | ' (5.21%) | _net (0.94%) | ' (7.68%) | _called (6.27%) | _simple (6.34%) | haus (5.11%) | 3 (6.41%) |
| Block 33 (3.65%) | , (2.08%) | ' (0.83%) | _assembly (5.90%) | ' (1.61%) | _to (5.86%) | _TW (1.51%) | Global (5.96%) | _LL (5.41%) |
| Block 34 (2.57%) | , (2.10%) | _to (1.01%) | _vide (0.99%) | , (2.72%) | _and (1.15%) | _class (1.00%) | lc (5.89%) | , (5.73%) |
| Block 35 (1.67%) | , (2.12%) | client (1.09%) | _NET (1.00%) | C (3.33%) | _and (2.74%) | _reservoir (1.16%) | Draft (1.02%) | _scripts (0.93%) |
| Block 36 (1.28%) | ¢ (2.69%) | C (1.06%) | gil (1.03%) | C (1.15%) | C (1.01%) | _Leopard (1.22%) | artist (1.05%) | stals (1.02%) |
| Logits | _ | " | _network | ' | _which | _neural | c | , |
| Expan. (0.980) | , | - | _network | _for | _which | _neural | - | , |

Figure 16: Joint jet lens with learnable weightings ($k = 0$), applied over GPT-2-large with the input sentence "new simple neural architecture, the Transformer"

| | new | _simple | _neural | _architecture | , | _the | _Trans | former |
|---|---|---|---|---|---|---|---|---|
| Block 1 (3.50%) | bie (3.17%) | * (4.75%) | _network (5.93%) | " (3.61%) | _which (1.15%) | _neural (1.60%) | c (5.06%) | _is (2.74%) |
| Block 2 (3.14%) | _ (0.84%) | * (4.15%) | _network (5.49%) | ' (1.80%) | _which (4.28%) | _neural (4.04%) | c (3.60%) | _is (0.93%) |
| Block 3 (1.19%) | _ (0.86%) | * (0.91%) | _network (0.84%) | ' (1.05%) | _which (1.81%) | _neural (2.17%) | c (0.78%) | _is (1.08%) |
| Block 4 (1.08%) | - (0.77%) | ton (1.88%) | _network (1.27%) | ' (0.99%) | _we (0.96%) | _neural (0.94%) | c (0.75%) | _is (1.07%) |
| Block 5 (0.98%) | _ (0.74%) | * (1.03%) | _network (0.98%) | ' (1.06%) | _where (1.01%) | _brain (1.00%) | c (0.88%) | _is (1.13%) |
| Block 6 (1.29%) | _ (3.29%) | * (1.01%) | _network (0.93%) | ' (1.07%) | _and (1.00%) | _neural (1.00%) | c (0.93%) | _is (1.06%) |
| Block 7 (1.32%) | _ (3.60%) | * (1.04%) | _network (0.97%) | ' (1.10%) | _which (1.00%) | parent (0.89%) | ient (0.97%) | _is (0.97%) |
| Block 8 (1.35%) | _ (3.71%) | * (1.05%) | _network (0.95%) | ' (1.07%) | _which (0.98%) | _researchers (0.99%) | iient (0.97%) | _is (1.10%) |
| Block 9 (1.44%) | , (3.74%) | * (1.04%) | _network (0.83%) | ' (1.07%) | _which (0.99%) | _neural (0.99%) | c (0.94%) | _is (1.91%) |
| Block 10 (1.47%) | - (3.73%) | * (1.04%) | _network (1.44%) | ' (1.07%) | _which (0.97%) | _neural (0.99%) | former (0.93%) | _AI (1.57%) |
| Block 11 (1.36%) | - (3.71%) | * (0.98%) | _network (1.01%) | ' (1.12%) | _which (0.98%) | _neural (0.98%) | c (0.99%) | _is (1.10%) |
| Block 12 (1.36%) | _ (3.69%) | * (1.00%) | _network (1.04%) | ' (1.08%) | _which (0.97%) | _neural (0.97%) | c (1.03%) | , (1.12%) |
| Block 13 (1.35%) | _ (3.65%) | * (1.01%) | _network (1.04%) | " (1.10%) | _where (0.96%) | _neural (0.96%) | c (1.01%) | _Cortex (1.09%) |
| Block 14 (1.31%) | _ (3.61%) | * (1.00%) | _network (1.02%) | ' (1.07%) | _a (0.74%) | _neural (0.92%) | ient (1.00%) | _is (1.10%) |
| Block 15 (1.30%) | _ (3.54%) | * (0.99%) | _network (1.03%) | ' (1.07%) | _which (0.93%) | _neural (0.93%) | c (1.00%) | _chip (0.90%) |
| Block 16 (1.30%) | _ (3.43%) | * (1.04%) | _network (0.95%) | ' (1.09%) | _and (0.89%) | _neural (0.89%) | c (0.99%) | , (1.13%) |
| Block 17 (1.28%) | _ (3.36%) | * (0.97%) | _network (0.95%) | ' (1.09%) | _which (0.90%) | _neural (0.86%) | c (0.99%) | . (1.10%) |
| Block 18 (1.14%) | _ (2.81%) | _ (0.92%) | _network (1.00%) | ' (0.90%) | _a (0.74%) | _more (0.79%) | c (0.90%) | _chip (1.09%) |
| Block 19 (0.99%) | _ (0.98%) | * (0.84%) | _network (0.88%) | ' (0.95%) | _or (1.44%) | _neural (0.76%) | c (0.98%) | _architecture (1.10%) |
| Block 20 (1.53%) | , (0.95%) | x (0.88%) | _network (0.95%) | ' (0.99%) | _we (3.52%) | _authors (3.11%) | c (0.77%) | _is (1.07%) |
| Block 21 (1.23%) | , (0.96%) | * (0.86%) | _networks (0.90%) | ' (1.04%) | _neural (1.93%) | _network (1.16%) | c (1.93%) | _is (1.07%) |
| Block 22 (1.92%) | - (0.96%) | * (2.47%) | _network (0.88%) | ' (1.05%) | _we (4.10%) | _neural (4.13%) | c (0.78%) | _Brain (0.98%) |
| Block 23 (2.10%) | _ (0.90%) | _stuff (0.79%) | _network (1.16%) | ' (0.85%) | _similar (3.67%) | _cu (4.65%) | c (3.79%) | _is (0.99%) |
| Block 24 (3.00%) | _ (0.93%) | * (2.25%) | _network (4.69%) | ' (2.88%) | ' (4.60%) | _ART (4.85%) | c (2.96%) | , (0.85%) |
| Block 25 (3.99%) | "]=> (3.39%) | ton (4.25%) | _net (2.85%) | ' (2.19%) | _with (4.38%) | _loc (4.88%) | c (5.43%) | _S (4.59%) |
| Block 26 (3.96%) | Instance (3.52%) | ' (3.67%) | _network (3.98%) | ' (4.45%) | _Cooper (4.93%) | _first (4.80%) | c (4.25%) | , (2.07%) |
| Block 27 (4.99%) | _ (3.24%) | tons (5.87%) | _network (4.56%) | _of (5.90%) | _but (4.78%) | _neuron (4.83%) | c (4.85%) | _Memory (5.85%) |
| Block 28 (5.13%) | _ (3.08%) | ton (5.20%) | _network (5.48%) | _for (5.93%) | _NI (4.98%) | _first (4.92%) | ient (5.17%) | _uses (6.28%) |
| Block 29 (5.04%) | _ (5.80%) | me (5.80%) | _network (5.64%) | *, (5.22%) | _NAT (4.95%) | _authors (4.94%) | ient (5.52%) | _3000 (5.00%) |
| Block 30 (4.88%) | _ (3.40%) | _kitchen (4.88%) | _network (5.69%) | " (5.41%) | _prototyp (4.94%) | _algorithm (4.88%) | ient (5.55%) | _uses (4.30%) |
| Block 31 (5.31%) | _ (3.61%) | x (6.06%) | _network (3.85%) | " (6.79%) | _geared (5.16%) | _traditional (5.00%) | c (5.28%) | _XL (6.76%) |
| Block 32 (5.51%) | - (3.70%) | _white (5.66%) | _network (5.56%) | " (6.48%) | ", (5.09%) | _WS (5.03%) | c (5.33%) | _is (7.26%) |
| Block 33 (5.75%) | , (3.73%) | ' (6.05%) | _network (6.01%) | " (6.91%) | _which (5.15%) | _neural (5.05%) | c (5.66%) | _Robot (7.46%) |
| Block 34 (5.88%) | , (3.73%) | ton (6.26%) | _network (6.49%) | *, (6.91%) | _which (5.15%) | _neural (5.04%) | ient (5.96%) | _Cortex (7.50%) |
| Block 35 (5.77%) | - (3.74%) | * (6.11%) | _network (6.26%) | _model (6.90%) | _modeled (5.03%) | _neural (4.97%) | ient (6.03%) | _model (7.17%) |
| Block 36 (5.85%) | _ (3.67%) | * (6.29%) | _network (6.51%) | ' (6.77%) | _which (4.95%) | _neural (5.00%) | c (6.10%) | _is (7.52%) |
| Logits | _ | " | _network | ' | _which | _neural | c | , |
| Expan. (0.994) | _ | " | _network | ' | _and | _neural | c | _is |

Figure 17: Joint jet lens with learnable weightings ($k = 1$), applied over GPT-2-large with the input sentence "new simple neural architecture, the Transformer"

Table 5: Several attention heads in the first residual block of *OLMo-7B* and their roles identified with jet tri-grams extracted from corresponding jet paths. We also include an example tri-gram captured by each head.

| Head Index | 2 | 16 | 26 | 30 |
|---|---|---|---|---|
| Role | Math/LaTeX | "for ... purposes" | date composition | "into account/consideration ..." |
| Example 3-gram | (␣Lemma, ␣let, ␣s) | (␣for, ␣use, ␣purposes) | (20, 23, ␣-) | (␣into, ␣account, ␣possible) |
| $\Delta$logit after intervention | $-0.1570$ | $-0.0019$ | $-0.0093$ | $-0.0001$ |

## G.2 ADDITIONAL TABLES OF JET PATHS OF INDIVIDUAL COMPONENTS

Table 5 reports a role identification study on attention heads in the first self-attention of *OLMo-7B* using jet tri-grams. Specifically, we find heads associated with math and programming, e.g. head 1 on Math/Latex; heads promoting digits and dash composition into dates, e.g. head 25; and heads constituting phrase templates, e.g. head 15 managing a "for $x$ purposes", where $x$ is a placeholder. To verify the roles we revealed, we further perform preliminary intervention experiments where we ablate MLPs or attention heads and compute variations in model logits. After the interventions, the logits drop consistently in all cases, suggesting our jet $n$-grams indeed can help identify certain roles for selected components. Varying impact on logit differences is likely due to overdetermination (Mueller, 2024) and our partial selection of jet paths (e.g. for tri-grams we only selected encoding-attention-decoding paths, excluding any MLP).

## H    CASE STUDY 2: ASSESSING FINETUNING EFFECTS (ADDITIONAL RESULTS ON RLHF TOXICITY AND REFUSAL MASS)

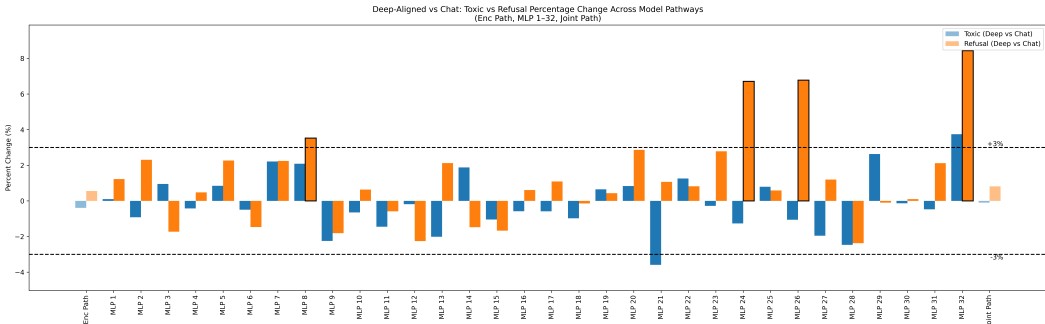

Figure 18: Refusal v.s Toxic bi-gram mass distribution across MLP paths.

A natural question is whether the toxic bigram mass changes significantly in the more robustly tuned models introduced in (Qi et al.). We thank the reviewers for the reference and the suggestion of this experiment. Our prior findings suggest that existing alignment methods tend to mask toxicity rather than eliminate it at the representation level. Here, we further investigate this phenomenon. Figure 18 summarize the results of Experiment 1 and Experiment 2 below. The orange bars indicate the refusal mass, while the blue bars indicate the toxic mass. For each layer, percentage change is computed as: $\frac{M_{\text{aug}} - M_{\text{chat}}}{M_{\text{chat}}}$, where $M_{\text{aug}}$ and $M_{\text{chat}}$ denote the bigram mass from the augmented and chat models, respectively. Positive values indicate an increase in mass.

**Experiment 1: Toxic Bigram Mass.** We compare `Llama2-7B-Chat-Augmented` (the open-sourced weights from Qi et al.) with the RLHF model `Llama2-7B-Chat`. Toxic bigram mass is measured in two ways: (1) independently for each Enc-(MLP)-Dec path (P1), and (2) jointly by averaging contributions across paths (P2), which is the metric reported in our main submission.

Under the joint metric (P2), we observe no substantial change in total toxic bigram mass. The augmented model reduces toxic mass by only 0.05% compared to `Llama2-7B-Chat`.

We further analyze the distribution of toxic mass across individual MLP blocks. For most layers, the relative change ($\frac{M_{\text{aug}} - M_{\text{chat}}}{M_{\text{chat}}}$), falls between $-0.5\%$ and $-2\%$, with only a few layers exhibiting reductions greater than $3\%$. These results suggest that the augmented model does not dramatically *reduce* toxic bigram mass, but instead reshapes its block-wise distribution.

**Experiment 2: Refusal Mass.** If the additional fine-tuning does not substantially reduce toxic mass, what accounts for the marked behavioral change (i.e., safer behavior appearing deeper into the response in Llama2-7B-Chat-Augmented)? We investigate whether behavioral differences arise from changes in refusal-related patterns. Using the same pipeline, we compute bigram mass for refusal phrases (textual patterns commonly used to reject unsafe queries) (Jain et al.; Arditi et al., 2024). We find that `Llama2-7B-Chat-Augmented` exhibits increased refusal mass in mid-to-late MLP layers relative to `Llama2-7B-Chat`. Notably, the final layer (MLP 32) shows a $+8.43\%$ increase in refusal mass, likely reflecting its proximity to the output layer.

**Discussion.** Together, these results suggest that the additional fine-tuning does not directly subtract toxic knowledge mass. Instead, it appears to strengthen representations of refusal phrases, thereby shifting toxicity-related behavior deeper into the generation process.

This interpretation aligns with the fine-tuning procedure described in (Qi et al.), which augments the corpus with examples containing "Safety Recovery" segments embedded within responses. We speculate that the observed increase in refusal mass emerges directly from this augmentation process, while the modest decrease in toxic mass may be an indirect consequence of adapting model statistics to the new data. Overall, the augmentation procedure effectively alters model behavior, but likely through **reinforcement of refusal patterns rather than explicit removal of toxic knowledge**.

## I  CASE STUDY 3: TRACING PRETRAINING DYNAMICS WITH JET BI-GRAMS

Pretraining an LLM is usually extremely resource intensive. Therefore it is crucial to monitor the progress of a pretraining run to prevent wasting of time and compute. In this section, we show how jet bi-grams can serve as an effective signaling tool to trace the pretraining dynamics, providing insights about the model's maturity. Such signals are especially useful to understand what happens with the model when the pretraining loss shows marginal improvements and fails to reflect the changes inside the model.

**Identifying the top bi-grams.**  To assess the model's progression, we extracted jet bi-grams from *OLMo-7B* model checkpoints across 555K pretraining steps. Table 6 presents a summary of the top 10 jet bi-grams at different stages of training. Due to space reason, we only show the top 10 jet bi-grams every 100K steps. Initially, the network exhibits nonsensical jet bi-grams, such as "`ICUirling`". As training advances, it gradually learns more meaningful combinations, like "`at least`". This process of acquiring sensible bi-grams stabilizes around step 200K, indicating that the model is reaching a level of maturity where the top 10 bi-grams capture common meaning.

**Learning speed.**  To evaluate the learning speed of jet bi-grams during pretraining, we consider the jet bi-grams at the final training step (555K) as the ground-truth bi-grams. We then chart the hit ratios of these ground-truth bi-grams at each pretraining step, as illustrated in Figure 20a. Interestingly, even though the pretraining loss (the blue curve) shows only minor improvements after the initial 50K steps, the model's acquisition of effective bi-grams continues to progress in a steady, consistent manner. Hence bi-grams learning dynamics are active throughout the training procedure, even after the training loss stabilizes. This indicates that there is significant behavior change in the model which is not well captured by the training loss, an observation that is studied also in grokking and double-descent (Zhang et al., 2021; Power et al., 2022). In other words, jet bi-grams may offer another point of view for analyzing the learning dynamics compared to pretraining loss. In addition, fig. 20b characterizes the total pseudo-joint probability mass of top 1K bi-grams from empirical data (Liu et al., 2024). We derive a pseudo-joint jet bi-gram probability using statistical uni-grams from (Liu et al., 2024). We observe that the model gradually accumulates probability mass that aligns with the real corpus data distribution.

**Learning schemes for different bi-grams.**  To understand if there are any differences between the learning schemes of different bi-grams, we can trace the progression of the jet bi-gram scores for selected bi-grams. Figure 19 provides a visual comparison of how different bi-grams are promoted or suppressed during the pretraining process. The different slopes and levels of the lines indicate varying rates of learning for the respective bi-grams. We observe that, the model first acquires random bi-grams due to random parameter initialization. These random bi-grams, like "`ICUirling`" and "`VENT thanks`", are quickly suppressed in the early steps and never regain high scores. In contrast, one-to-many bi-grams like "`at least`" are first promoted to very high scores but then get suppressed perhaps due to the model seeing more of the scope of the token "at". One-to-one bi-grams like "`&amp`" (HTML code) are gradually promoted and stabilize. Many-to-many bi-grams like "`make sure`" takes the most time to learn and the scores are still increasing even at the end of pretraining. Our findings suggest that the training process effectively promotes certain "good" bi-grams, but at different paces, where they might be suppressed later depending on their occurrences and linguistic nature. These insights could inform future training strategies, such as targeted training on more relevant bi-grams or adjusting the training data to improve the pretraining speed.

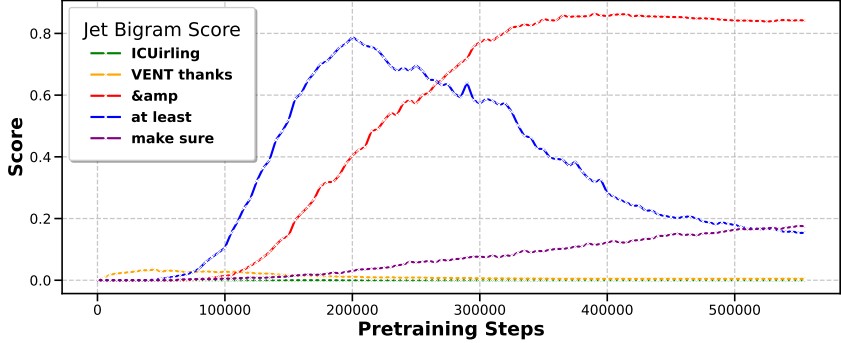

Figure 19: Visualization of *OLMo-7B*'s promotion and suppression dynamics of jet bi-grams scores.

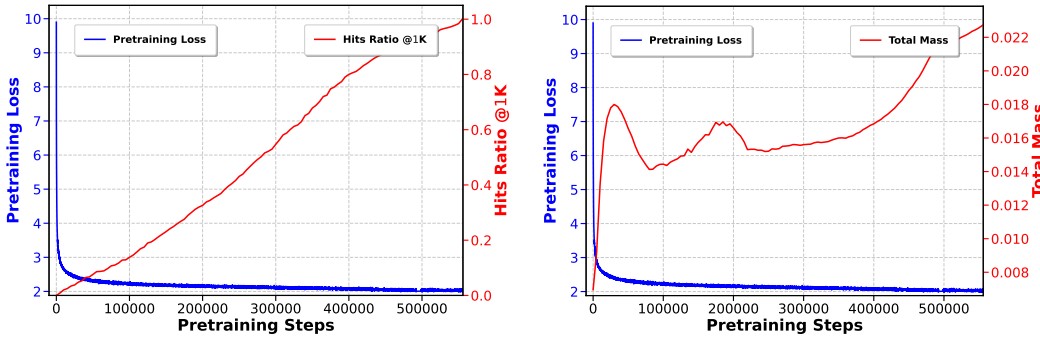

(a) Top 1K jet bi-gram hit ratios w.r.t. the final step.  (b) Top 1K jet bi-gram mass w.r.t. empirical data.

Figure 20: Analysis of *OLMo-7B*'s pretraining dynamics via measuring its jet bi-gram progression.

Table 6: Bi-gram evolution across pretraining steps for OLMo 7B. Each column represents a distinct step, while each row corresponds to a different rank. The table entries are the bi-grams at each step for each rank. The number of tokens seen in association with the pretraining steps is also annotated. The model gradually picks up meaningful bi-grams after starting from senseless bi-grams (due to random initialization).

| Rank | 0K [#steps] 0B [#tokens] | 100K 442B | 200K 885B | 300K 1327B | 400K 1769B | 555K 2455B |
|---|---|---|---|---|---|---|
| 0 | immortal | 's | at least | &amp | &amp | &amp |
| 1 | ICUirling | at least | 's | at least | its own | its own |
| 2 | ords architect | its own | &amp | its own | their own | their own |
| 3 | yaml Adam | okerly | your own | your own | at least | his own |
| 4 | 231 next | VENT thanks | its own | their own | your own | make sure |
| 5 | clonal条 | iums | iums | more than | his own | your own |
| 6 | Charg@{ | you're | you're | can't | 2nd | 2nd |
| 7 | avoir careless | Everything v | 2nd | his own | more than | at least |
| 8 | HOLD worsening | erna already | you guys | 2nd | make sure | more than |
| 9 | Horse dismant | 'my | more than | make sure | can't | iums |

