# Decomposing LLM Computation with Jets

**Yihong Chen[δ], Xiangxiang Xu[ℵ], Pontus Stenetorp[Υ], Sebastian Riedel[Υ], Luca Franceschi[Ω]**

[δ]OATML, University of Oxford, UK    [Υ]AI Centre, University College London, UK
[Ω]Independent researcher, Berlin, Germany    [ℵ]University of Rochester, USA

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

 = \text{Dec} \circ \Big( \underbrace{\text{Enc}}_{x_0} + \underbrace{\gamma_1 \circ \text{Enc}}_{x_1} + \underbrace{\gamma_2 \circ \big(\text{Enc} + \gamma_1 \circ \text{Enc}\big)}_{x_2} \Big).$$

The nested parentheses entangle contributions: the outer (purple) grouping mixes everything, while the inner (orange) ties $\gamma_2$ to both $x_0$ and $x_1$. Traditional MI would select paths syntactically, akin to selecting modules in a PyTorch computation graph, ignoring these nesting effects. Jets let us cut both levels systematically and isolate the contributions of different input→output paths.

**Step 1: Inner expansion.** At $\gamma_2$, taking $\{x_0, x_1\}$ as jet base points and using Lemma 1, the residual stream $x_2 = \gamma_2 \circ (x_0 + x_1)$ can be decomposed as

$$x_2 \approx_k J^k\gamma_2(x_0 + x_1) = \underbrace{w_0 J^k\gamma_2(x_0)}_{x_{20}} + \underbrace{w_1 J^k\gamma_2(x_1)}_{x_{21}} + O(r^{k+1}),$$

so the original entangled stream $x_2$ separates into two sub-streams, as illustrated in Figure 3(b).

**Step 2: Outer expansion.** At Dec, the jet base points are updated from $\{x_0, x_1, x_2\}$ to $\{x_0, x_1, x_{20}, x_{21}\}$ after previous expansion. Using Lemma 1 and jet algebra (Proposition 1), yields

$$f \approx_k J^k\text{Dec}(x_0 + x_1 + x_{20} + x_{21}) = \underbrace{\bar{w}_0 J^k(\text{Dec} \circ \text{Enc})}_{f_\emptyset} + \underbrace{\bar{w}_1 J^k(\text{Dec} \circ \gamma_1 \circ \text{Enc})}_{f_{\{1\}}}$$

$$+ \underbrace{\bar{w}_2 J^k(\text{Dec} \circ (w_0 J^k(\gamma_2 \circ \text{Enc})))}_{f_{\{2\}}} + \underbrace{\bar{w}_3 J^k(\text{Dec} \circ (w_1 J^k\gamma_2(\gamma_1 \circ \text{

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

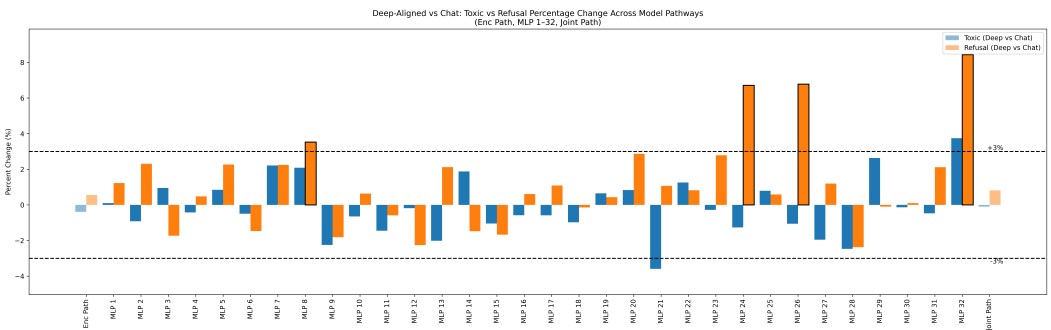

Figure 20: Refusal bi-gram mass distribution across MLP paths.