# OpenReview forum: "Decomposing LLM Computation with Jets"
_ICLR.cc/2026/Conference — ICLR 2026 Poster_

### Official Review · Reviewer_QKXz · 2025-10-21

**Soundness:** 3
**Presentation:** 2
**Contribution:** 3
**Rating:** 6
**Confidence:** 2

**Summary:**

The paper proposes a new interpretability method for examining transformer-based language models based on jet expansions. The method works by approximating the full language model by its Taylor expansion and analyzing low-order terms. This extracts low-order behavior from the model and allows one to interpret it functionally, without relying on any additional data or training. The authors propose an implementation of the framework for a transformer-based language model and analyze its efficiency.
Jet expansions subsume existing interpretability techniques such as logit lens and $n$-gram statistics-based interpretability methods. Thus, the authors use their method to analyze open-source language models and find interpretable results. For example, by comparing fine-tuned and non-fine-tuned models, they find that the fine-tuning did not affect certain parts of model behaviour.

**Strengths:**

- The proposed methodology seems broadly applicable and useful
	- I particularly like that the approach is dataset-free, which removes many possible confounders in the analysis
	- Similarly, I like that the method is more holistic than individual neurons, etc.
- The algorithm is thoroughly described and seems applicable and useful
	- It’s interesting that the analysis can even be done on CPUs
- I liked the thoroughness of the experiments, which cover many open-source models.
- Experiments clearly show that taking higher-order jets ($k = 1$ instead of $k = 0$ for logit lens) actually matters

**Weaknesses:**

- Unfortunately, I found it a bit hard to follow the transition between the empirical results and the theoretical setup. More explicitly stating what the measured quantities in the experiments correspond to in terms of the theory and the notation introduced above would be useful here, I think.
	- For example, how do $n$-gram statistic map to jets?
- It is unclear to me to what degree the results should be trusted as a truthful representation of the model. For example, without knowing anything in advance, is it not possible that most of the model behavior is only explained by higher order terms than those in the jet?
- The work mostly reframes existing techniques as jet expansions but doesn’t, to my understanding, propose novel applications of this framework for analyses that weren’t possible before.

**Questions:**

- Is there any way to understand how good the approximation is, i.e., how large the remainders are?
- How much of this methodology relies on the transformer architecture? Would it be easy to port to other neural LM architectures?
- Could the same methodology even be used for things like model compression (where only the jets are retained/used)?

---

> ### Author Response · Authors · 2025-11-26
>
> Thank you for reviewing our paper and recognizing the advantage of a dataset-free holistic interpretability framework that is applicable and useful even when run on CPU. Thanks also for appreciating the thoroughness of the experiments and recognizing the effectiveness of higher-order Jet lenses.
>
> We address your concerns as follows:
>
> ## W1 Exposition on the experiments
> > More explicitly stating what the measured quantities in the experiments correspond to in terms of the theory and the notation introduced above would be useful here, I think. For example, how do n-gram statistic map to jets?
>
> Thanks for the suggestion. In the submission, due to space reason, we compressed the n-gram description. We have now **revised** the description of n-grams, and further elaborated bi/tri-gram algorithmic details in Appendix E.
>
> For measured quantities in experiments, we have **added** description on experimental metrics in Appendix F, including choosing cosine similarity as a remainder metric, how to do jet bi-gram comparison, etc. We also **added** a paragraph connecting the theory to the empirical setup in Sec. 5.1. We hope this revision addresses your concerns.
>
> ## W2 model behavior explained by higher-order terms
> > It is unclear to me to what degree the results should be trusted as a truthful representation of the model. For example, without knowing anything in advance, is it not possible that most of the model behavior is only explained by higher order terms than those in the jet?
>
> Our use of jets is intended for interpretability, not for constructing an accurate surrogate of the model. Conceptually, we find it useful to compare jet expansions to Fourier transform: even when the Fourier transform captures only part of the signal’s spectrum and the resulting approximation error cannot be directly measured, the partial frequency information it provides can still be valuable for understanding the signal and diagnosing issues in the generating circuits.
>
> Although in principle some model behavior may be governed by higher-order structure, we expect that the framework yields most of the insightful information on lower k’s. This is because lower-order expansions are cheaper, smoother, and more stable numerically. In addition, there is some interpretability research (e.g., the Linear Representation Hypothesis [1]) showing that LLMs generally present a high degree of linearity in their representations.
>
> On the other hand, Taylor-based expansions may not necessarily be the best tool for recovering high-order behaviour (say, for k > 2). Indeed, nonlinearities such as layernorm introduce non-convergent or poorly conditioned higher-order terms. Empirically, going beyond k = 2 they likely produce unstable expansions.
>
> In general, the framework aims at revealing stable, legible portions of the computation (likely reflected in low-order parts) rather than to approximate all model behaviour (see also Q3 – compression). Exploring alternative bases that better capture higher-order structure is an important direction for future work.
>
> ## W3 novelty in applications
> > The work mostly reframes existing techniques as jet expansions but doesn’t, to my understanding, propose novel applications of this framework for analyses that weren’t possible before.
>
> We appreciate the concern. We believe that while a core contribution is unification and reframing, our framework does lead to fully novel applications. In particular, we believe that systematically extracting n-grams directly from the LLM (without further data nor training) is a novel tool, which, in turn, allows for novel applications such as analyzing fine-tuning effects (Sec. 5.2.2) and tracking training progress (Appendix G). This is a capability prior methods do not offer, also acknowledged by Reviewer `F5cx`, `tf63`, and `KiGr`.
>
> We also view the Jet lens, especially the joint version, as a new tool for instance-level interpretability backed by theoretical underpinning. And Sec. 5.2.1 illustrates that our jet lens are better than Logit Lens, which fails models like GPT-Neo. This advantage is also noted by Reviewer `F5cx`.
>
> ## Q1 Remainder size
> We apologize for the confusion and have **revised** the paper, splitting and expanding Remark 2 for clarity. In summary, the framework does not aim to provide approximation guarantees. Instead, in the following we clarify when small remainders are expected (input-specific cases) and when they are not (function-level expansions), but the expansions do still yield meaningful, interpretable insights.
>
> In **input-specific evaluations** (where we choose specific input sentences like “new simple neural architecture, the Transformer”), we expect remainders to be small, since we want to draw specific conclusions about how a model is behaving on a particular input. In these cases (i.e., in the Jet lenses part of the experiments) we do provide empirical studies, and we often find that the remainder δ is small. (to be continued)

---

> ### Author Response · Authors · 2025-11-26
>
> (continuing) In order to measure the remainder between the expansion logit and the model logit, we compute the **cosine similarity** between them, since direct difference depends heavily on the input sentences (See the **revised** Appendix F, “Cosine similarity as a remainder metric.”). In short, we use cosine similarity as an indicator for checking if the remainder is small. Most of our heatmaps in the submission include this similarity measure in parentheses, e.g. *"Expan. (0.993)"*. We also compute average similarities over 100 examples and summarize the results in Fig. 4 (bottom): expansion logits are frequently close to the model outputs, with cosine similarities in the 0.85–1 range, indicating that the extracted components capture most of the behavior for many concrete inputs, i.e. smaller remainders.
>
> In general, the remainder size depends on three factors:
> 1. the type of jet expansion one performs (e.g., iterative vs joint);
> 2. the non-linearity on which the jet is applied (e.g., ReLU vs ELU vs LayerNorm);
> 3. how far the variate is from the center point, which in turn depends fully on the chosen input sentence (see Eq. 4).
>
> In **function-level expansions** (e.g., around the embedding function for extracting bi-grams), remainders can be large and this is fully expected. Here, the goal is not approximation accuracy but to **decompose the computation into interpretable paths**.
>
> Concretely, consider bi-grams (Sec. 5.2.2): the model performs far more than the isolated mechanism of predicting a token based solely on the previous one. Therefore, we *do* expect remainders between the model output and the extracted bi-gram paths to be naturally large. Yet the extracted paths remain meaningful precisely because they isolate a coherent part of the computation, even if they explain only a small fraction of the total behavior. A large remainder therefore does not invalidate the interpretation; it simply reflects that we are focusing on one specific, coherent part of the computation. This is a common situation across interpretability research.
>
> Conceptually, we find it useful to compare jet expansions to a Fourier transform: even when the Fourier transform captures only part of the signal’s spectrum and the resulting approximation error cannot be directly measured, the partial frequency information it provides can still be valuable for understanding the signal and diagnosing issues in the generating circuits.
>
> We have **added** a discussion in the revised Appendix B. We apologize if this was unclear in the original submission and hope that this addresses your concerns.
>
> ## Q2 Architecture
> Our theory mainly relies on the residual computational graph. It would be straightforward to port to neural LMs (or other modalities e.g. ViT) that follows encoder-residual block $\times L$ - decoder framework.
>
> ## Q3 Model compression
> > Could the same methodology even be used for things like model compression (where only the jets are retained/used)?
>
> We thank the reviewer for raising this interesting possibility.
>
> Our aim when developing this has been quite the opposite of model compression. We see Jet expansions acting as decompressors: they take a highly compact, monolithic neural LLM and expand it into explicit input–output paths, each of which can be easier for interpreting. From this perspective, modern neural LLMs are already extremely compressed compared to classical ($|V|^{W}$) n-gram language models ($|V|$ is vocabulary size, $W$ is the window size), and jet expansions serve to unfold that compressed representation into a more transparent form such as jet n-grams.
>
> In general, the jet-expanded representation is not designed to reduce parameter count, inference latency, or memory footprint, which the line of compression work focuses on. The purpose of using jets is analytical, not operational.
>
> That said, the reviewer’s idea points to interesting future directions. Low-order jets could potentially be used for constructing simplified surrogate models that capture broad trends across prompts. These would not constitute “compression’’ in the classical sense but might provide ways to derive compact approximators inspired by jet structure. Exploring such possibilities is outside the scope of this paper and our expertise, but could be a promising direction for future work.
>
> **Reference**
>
> [1] Park, Kiho, Yo Joong Choe, and Victor Veitch. "The Linear Representation Hypothesis and the Geometry of Large Language Models." International Conference on Machine Learning. PMLR, 2024.

---

> > ### Comment · Reviewer_QKXz · 2025-11-27
> >
> > Dear authors,
> >
> > Thank you very much for taking the time to thoroughly address my questions. The answers and revisions to the paper have indeed helped a lot, and I now understand the contributions much better. I have adjusted my evaluation, and I hope the paper gets accepted.

---

> > > ### Author Response · Authors · 2025-11-28
> > >
> > > Thank you for your prompt response amid the chaos. It appears that OpenReview currently doesn’t allow score changes at the moment, but we appreciate your intention and your curious, constructive feedback throughout the process.

---

### Official Review · Reviewer_KiGr · 2025-11-01

**Soundness:** 4
**Presentation:** 4
**Contribution:** 3
**Rating:** 8
**Confidence:** 3

**Summary:**

- The authors introduce Jet Expansions, a framework for expanding computational graphs using jet operators that give interpretability to complex models (e.g., transformers).
- A jet expansion uses a transformer's residual structure to expand it to many smaller layer combination terms, plus some remainder.
- For interpretability, you look at the different orders of jets to understand what the model is trying to predict.

**Strengths:**

- The paper's main contribution, turning LLMs into jet-expanded paths, requires only model access and no extra data to probe models or compare two models with a shared vocabulary.
- The paper does a good job explaining the background and motivation, given the complexity of the work.

**Weaknesses:**

I question the practicality of jet expansions for interpretability. The paper only demonstrates the method on small to mid-sized open models, and even there, higher-order jets already add significant runtime. It's unclear how feasible this is for larger / frontier models.

**Questions:**

How large are remainders in practice, and can we be more explicit about their sizes?

---

> ### Author Response · Authors · 2025-11-26
>
> Thanks for reviewing our paper and for appreciating the advantages of sidestepping probing datasets and the clarity of the background and motivations. We address your concerns as follows.
>
> ## W1 larger / frontier models
> > I question the practicality of jet expansions for interpretability. The paper only demonstrates the method on small to mid-sized open models, and even there, higher-order jets already add significant runtime. It's unclear how feasible this is for larger / frontier models.
>
> We appreciate your concern. First of all, we want to highlight that our framework requires open-weights models. In this work we managed to run with our computational resources experiments on jet bi-grams for Llama 2 70B. We did not have time to look deeply into the jet bi-gram difference between Llama 2 7B and Llama 2 70B. We will include the jet bi-grams for Llama 2 70B in the appendix.
>
> In general, we believe that with more resources Jet expansions can be applied to larger models, at least for k=0 or 1. In particular:
>
> - With **uniform weights**, expansions with **k = 0** are in the same runtime/memory order as forward passes to the model (meaning that whenever you are able to make forward passes, you should be able to compute k = 0 Jet expansions).
> - **k = 1** expansions are on the same order as gradients propagation, meaning that whenever you can compute gradients, you can compute k = 1 Jet expansions.
> - **k = 2** expansions require Hessian–vector products, which we acknowledge would be too expensive to compute for the largest models.
>
> Nevertheless, as we discussed throughout, we believe **k = 0 or 1 expansions already yield valuable insights**, and, practically, going beyond k = 2 is not necessary.
>
>
> ## Q1 remainder size
> > How large are remainders in practice, and can we be more explicit about their sizes?
>
> We apologize for the confusion and have **revised** the paper, splitting and expanding Remark 2 for clarity. In summary, the framework does not aim to provide approximation guarantees. Instead, in the following we clarify when small remainders are expected (input-specific cases) and when they are not (function-level expansions), but the expansions do still yield meaningful, interpretable insights.
>
> In **input-specific evaluations** (where we choose specific input sentences like “new simple neural architecture, the Transformer”), we expect remainders to be small, since we want to draw specific conclusions about how a model is behaving on a particular input. In these cases (i.e., in the Jet lenses part of the experiments) we do provide empirical studies, and we often find that the remainder δ is small.
>
> In order to measure the remainder between the expansion logit and the model logit, we compute the **cosine similarity** between them, since direct difference depends heavily on the input sentences (See the **revised** Appendix F, “Cosine similarity as a remainder metric.”). In short, we use cosine similarity as an indicator for checking if the remainder is small. Most of our heatmaps in the submission include this similarity measure in parentheses, e.g. *"Expan. (0.993)"*. We also compute average similarities over 100 examples and summarize the results in Fig. 4 (bottom): expansion logits are frequently close to the model outputs, with cosine similarities in the 0.85–1 range, indicating that the extracted components capture most of the behavior for many concrete inputs, i.e. smaller remainders.
>
> In general, the remainder size depends on three factors:
> 1. the type of jet expansion one performs (e.g., iterative vs joint);
> 2. the non-linearity on which the jet is applied (e.g., ReLU vs ELU vs LayerNorm);
> 3. how far the variate is from the center point, which in turn depends fully on the chosen input sentence (see Eq. 4).
>
> In **function-level expansions** (e.g., around the embedding function for extracting bi-grams), remainders can be large and this is fully expected. Here, the goal is not approximation accuracy but to **decompose the computation into interpretable paths**.
>
> Concretely, consider bi-grams (Sec. 5.2.2): the model performs far more than the isolated mechanism of predicting a token based solely on the previous one. Therefore, we *do* expect remainders between the model output and the extracted bi-gram paths to be naturally large. Yet the extracted paths remain meaningful precisely because they isolate a coherent part of the computation, even if they explain only a small fraction of the total behavior. A large remainder therefore does not invalidate the interpretation; it simply reflects that we are focusing on one specific, coherent part of the computation. This is a common situation across interpretability research. (to be continued)

---

> ### Author Response · Authors · 2025-11-26
>
> (continuing)
>
> Conceptually, we find it useful to compare jet expansions to a Fourier transform: even when the Fourier transform captures only part of the signal’s spectrum and the resulting approximation error cannot be directly measured, the partial frequency information it provides can still be valuable for understanding the signal and diagnosing issues in the generating circuits.
>
> We split Remark 2 in the paper to clarify things, and added a discussion in Appendix B. We apologize if this was unclear in the original submission and hope that this addresses your concerns.

---

### Official Review · Reviewer_tf63 · 2025-11-01

**Soundness:** 2
**Presentation:** 3
**Contribution:** 3
**Rating:** 6
**Confidence:** 2

**Summary:**

This paper introduces Jet Expansions, a framework that expands computational graphs using jet operators (generalizing truncated Taylor series) to decompose transformers into input-output computational paths and complementary remainders. The authors demonstrate how this functional decomposition stimulates advances in the area of interpretability applications, such as grounding and subsuming the traditional interpretability techniques such as the Logit Lens technique and defending against jailbreak attacks.

**Strengths:**

1. The paper has an impressive theoretical foundation. They introduced the concept of jet operators from differential geometry to define interpretability rigorously, which is both novel and effective.

2. Using the novel framework, the paper elegantly unifies the framework and as a result, introduces a technique to subsume traditional foundations of interpretability.

3. Unlike most interpretability methods, jet expansions operate directly on model structure without requiring probe datasets. This is a significant practical advantage.

4. The topic has broad applications, including model safety and mechanistic analysis

**Weaknesses:**

My major concern is the remainder interpretation. The remainder $\delta$ seems to be a critical quantity to ensure performance and credibility, yet it is not analyzed rigorously enough. For instance, in Remark 2 on page 6, they mentioned "Empirically, remainders are often small and expansion logits nearly collinear with model outputs". From a theoretical point of view, how small is "small enough for the task"?

**Questions:**

Please see Weaknesses.

---

> ### Author Response · Authors · 2025-11-26
>
> Thanks for reviewing our paper and for recognizing the value of the theoretical foundation of our work, the novelty, the elegant unification, the practical advantages of sidestepping probing datasets, and the broad applications of our proposed framework.
>
> We would like to address your concerns as follows.
>
> ## W1 & Q1 Remainder
> > My major concern is the remainder interpretation… From a theoretical point of view, how small is “small enough for the task”?
>
> We apologize for the confusion and have **revised** the paper, splitting and expanding Remark 2 for clarity. In summary, the framework does not aim to provide approximation guarantees. Instead, in the following we clarify when small remainders are expected (input-specific cases) and when they are not (function-level expansions), but the expansions do still yield meaningful, interpretable insights.
>
> In **input-specific evaluations** (where we choose specific input sentences like “new simple neural architecture, the Transformer”), we expect remainders to be small, since we want to draw specific conclusions about how a model is behaving on a particular input. In these cases (i.e., in the Jet lenses part of the experiments) we do provide empirical studies, and we often find that the remainder δ is small.
>
> In order to measure the remainder between the expansion logit and the model logit, we compute the cosine similarity between them, since direct difference depends heavily on the input sentences (See the **revised** Appendix F, “Cosine similarity as a remainder metric.”). In short, we use cosine similarity as an indicator for checking if the remainder is small. Most of our heatmaps in the submission include this similarity measure in parentheses, e.g. *"Expan. (0.993)"*. We also compute average similarities over 100 examples and summarize the results in Fig. 4 (bottom): expansion logits are frequently close to the model outputs, with cosine similarities in the 0.85–1 range, indicating that the extracted components capture most of the behavior for many concrete inputs, i.e. smaller remainders.
>
> In general, the remainder size depends on three factors:
> 1. the type of jet expansion one performs (e.g., iterative vs joint);
> 2. the non-linearity on which the jet is applied (e.g., ReLU vs ELU vs LayerNorm);
> 3. how far the variate is from the center point, which in turn depends fully on the chosen input sentence (see Eq. 4).
>
> In **function-level expansions** (e.g., around the embedding function for extracting bi-grams), remainders can be large and this is fully expected. Here, the goal is not approximation accuracy but to **decompose the computation into interpretable paths**.
>
> Concretely, consider bi-grams (Sec. 5.2.2): the model performs far more than the isolated mechanism of predicting a token based solely on the previous one. Therefore, we *do* expect remainders between the model output and the extracted bi-gram paths to be naturally large. Yet the extracted paths remain meaningful precisely because they isolate a coherent part of the computation, even if they explain only a small fraction of the total behavior. A large remainder therefore does not invalidate the interpretation; it simply reflects that we are focusing on one specific, coherent part of the computation. This is a common situation across interpretability research.
>
> Conceptually, we find it useful to compare jet expansions to a Fourier transform: even when the Fourier transform captures only part of the signal’s spectrum and the resulting approximation error cannot be directly measured, the partial frequency information it provides can still be valuable for understanding the signal and diagnosing issues in the generating circuits.
>
> We have **revised** Remark 2 by splitting it into two in the paper, and added a discussion in Appendix B. We apologize if this was unclear in the original submission and hope that this addresses your concerns.

---

### Official Review · Reviewer_F5cx · 2025-11-01

**Soundness:** 4
**Presentation:** 3
**Contribution:** 4
**Rating:** 8
**Confidence:** 3

**Summary:**

The authors aim to solve the problem of "entangled" computation in LLMs, which makes their internal workings difficult to interpret. They use the core insight to treat LLMs as residual networks (a sum of paths) and apply jet operators, the functional equivalent of Taylor series, to analyze them. This approach is well-suited for the problem because jets are mathematically designed to handle compositions of functions and, via Lemma 1, can "disentangle" a nonlinear function of a sum into a weighted sum of its parts. This functional decomposition allows the authors to "carve out" and analyze specific computational paths. Their main contributions are the "jet lens," which generalizes the popular Logit Lens (showing it's just the $k=0$ case), and "jet n-grams," a novel "corpus-free" method to extract symbolic knowledge tables directly from the model's parameters. The authors provide strong empirical results, showing their decompositions are highly faithful (high cosine similarity) and that their $k>0$ jet lens provides a more stable interpretation for models like GPT-Neo where the standard Logit Lens has more chaotic results.

**Strengths:**

Principled Framework: The paper's primary strength is its formalism. It gives interpretability a more formal mathematical framework/tool grounded in approximation theory.

Generalization of Existing Tools: Proving that the Logit Lens is simply the $k=0$ case of the iterative jet lens is a great benfit. It places a popular tool on solid theoretical footing and simultaneously shows its limitations.

Empirical Support: The results strongly support the claims.

Faithfulness: The high cosine similarity (e.g., 0.993 in Figure 4) between the expansion logits and the model logits shows the decomposition is not just theoretical but empirically sound.

Superiority: The $k>0$ iterative lens provides a much more stable and interpretable computational trace for models like GPT-Neo, where the standard $k=0$ Logit Lens is more chaotic (as shown clearly in Appendix I, Figs. 8 vs. 9).

Applications & Insights:

RLHF & Toxicity: The finding in Table 4 provides strong, quantitative evidence that RLHF alignment masks toxic associations rather than erasing them (ToxiGen score drops to 0, but toxic bi-gram mass is unchanged). The method of demonstrating it via a "corpus-free" n-gram analysis is novel and powerful.

"Corpus-Free" n-grams: The idea of extracting symbolic n-gram tables without a dataset is a powerful one. The application of "diffing" models (Llama-2 vs. CodeLlama) by comparing their n-gram tables is a practical method for verifying fine-tuning.

Future Work: The framework opens up many exciting possibilities for "functional-level" interpretability.

**Weaknesses:**

Tractability and Clarity: The paper's discussion of the $O(2^L)$ exponential expansion (Algorithm 2) versus the practical, $O(L)$ linear-in-depth applications (the lenses) could be clearer. The tractability of these expansions, especially for $k>1$ is a major practical concern that is only briefly touched upon.

New Hyperparameters: The method introduces new hyperparameters for interpretability, namely the jet order $k$ and the jet weights $w$. The sensitivity of the results to these choices is not fully explored. Also how efficient and practical is it to get arbitrary order expansions?

**Questions:**

The RLHF & toxicity result supports some results from "Safety Alignment Should be Made More Than Just a Few Tokens Deep" that alignment usually only masks the problems. It would be interesting to see if the toxic bi-gram mass changed significantly in the "more robustly" tuned models from "Safety Alignment Should be Made More Than Just a Few Tokens Deep".

Cost of $k>0$ Lenses: Could the authors quantify the practical computational overhead of using the $k=1$ iterative jet lens (Fig. 9) versus the $k=0$ lens (Fig. 8)? How much more expensive is it, and does this limit its real-world applicability as a drop-in replacement?

Choice of $k$: The results for GPT-Neo are dramatically better for $k=1$, but the results for GPT-2-large (Figs. 11 vs. 12) seem comparable. Does this suggest $k=0$ is "good enough" for some model families? How should a practitioner choose the optimal $k$ for a new model?

Would the authors release the code?

Notation in Example 1: The expression $J^1\gamma(x_1)(x_2)$ is unclear. Could the authors confirm this is meant to be $J^1\gamma(x_1)(x_1+x_2)$, as I think it expands as $J^1 \gamma(x_1)(x_1+x_2) = \gamma(x_1) + \gamma'(x_1)((x_1+x_2) - x_1)$?

Lemma 1 Proof: In the proof of Lemma 1 (Appendix A), the evaluation point for the jets appears to be mislabeled as 'x' when it should be 'y' (e.g., $\sum w_i J^k f(x_i)(x)$). Can the authors confirm if this is a typo? Or clarify the difference between x and y?


"Data-Free" Clarification: The "data-free" claim for n-grams seems to be a source of confusion. The method is clearly input-dependent (it evaluates on token embeddings). Would "corpus-free" or "dataset-free" be a more accurate descriptor?

---

> ### Author Response · Authors · 2025-11-26
>
> Thank you for your detailed, thoughtful, and constructive review.
>
> First of all,  we appreciate that you recognized the work's spirit of functional-level interpretability,  its attempt to offer a unifying theoretical grounding, the empirical soundness it strives for, the effectiveness of the jet lens, the practicality of the “corpus-free” n-gram approach, and other potentially broader possibilities opened by jet expansion.
>
> We now proceed to address your curious questions and related concerns.
>
> ## W1: Algorithm 2 exponential expansion.
> > The paper's discussion of the O(2^L) exponential expansion (Algorithm 2) versus the practical, O(L) linear-in-depth applications (the lenses) could be clearer. The tractability of these expansions, especially for k>1 is a major practical concern that is only briefly touched upon.
>
> We apologize for the confusion. We see the Jet expansions as a general framework that may have various instantiations, some of which are more theoretically-oriented (like the exponential expansion), some others are more practical, like the jet lenses and the jet n-grams.
>
> **Exponential expansion:** We wanted to include the exponential expansion in our paper because we believe it theoretically grounds previous work on "ensembling behaviour” of recursive residual architecture (which most LLMs and vision models, such as Veit et al 2016, adopt). We believe this line of investigation is today particularly relevant since the ensemble perspective on LLMs has been impacting techniques in various domains such as inference-time acceleration. A formal explanation as in Algorithm 2 provides theoretical backings on why LLMs are robust to runtime structure changes, such as Layerskip [0]; why dropout [6] is no longer essential in LLM training: they were kind of bringing the exponential ensemble effect [6-8], which coincide with the exponentially ensembling effect shown in Algorithm 2.
>
> **Tractability:** From a practical perspective, we believe that the expensive expansions as the exponential one could still be useful and applied (at least for k=0 or 1) by incorporating, for instance, sampling approaches (as for instance, Veit et al 2016 have done in their experiments). However, we leave these extensions to future work.
>
> In summary, by reframing observations in a functional formalism, we can revisit various earlier empirical findings like Veit et al 2016 with a principled analytical foundation and contextualize them with the latest progress of LLMs. We added a section in the appendix that touches on these topics (Appendix C).
>
>
> ## Q1 RLHF toxicity experiments
> > It would be interesting to see if the toxic bi-gram mass changed significantly in the "more robustly" tuned models from "Safety Alignment Should be Made More Than Just a Few Tokens Deep".
>
> Thank you for the reference and the suggestion of the new experiment. Indeed our existing findings support [1] in that existing alignment mainly mask the problem instead of rooting them out.
>
> **Experiment 1 Toxic Mass**
> We run the suggested experiment using Llama2-7B-Chat-Augmented, the open-sourced model weights from [1], and compare the result with the RLHF model, Llama2-7B-Chat, to check if their toxic bigram mass changes significantly. We first check the toxic bigram mass for each Enc-(MLP)-Dec path independently (P1) and also jointly (P2) (i.e. averaging up their contributions: this latter measurement is the one we reported in the submission). The findings are a bit beyond our expectation but we find it interesting and would like to share it here with you.
>
> First of all, we find no massive change in the total toxic bigram mass from P2. Llama2-7B-Chat-Augmented only reduces slightly (0.05%) more total toxic bigram mass compared to Llama2-7B-Chat. This makes us curious about what's changed in the additional data augmentation finetuning performed by [1].
>
> Then, we check the distribution of the bigram mass across different MLPs. When we compare Llama2-7B-Chat-Augmented with Llama2-7B-Chat, we find that most MLPs show toxic mass reduction between 0.5%–2% (the percentage is calculated by taking the ((augmented mass - chat mass) / chat mass) for each MLP). Only a few layers reduce more than 3%. This confirms Llama2-7B-Chat-Augmented does not dramatically reduce toxic bigram mass but seems to rather reshape the block-wise distribution. (to be continued)

---

> ### Author Response · Authors · 2025-11-26
>
> (continuing)
>
> **Experiment 2 Refusal Mass**
> If the additional fine-tuning does not reduce toxic mass dramatically, what made the model behavior change so much (safe behaviour is deeper into the response in Llama2-7B-Chat-Augmented)? To answer this question, we perform the same pipeline for retrieving bigram mass but this time with refusal phrases, a type of textual pattern that leads to refusing unsafe queries [2,3]. Interestingly, we see Llama2-7B-Chat-Augmented acquired more refusal mass in the mid to later MLPs, compared to Llama2-7B-Chat. Particularly, the MLP at the last layer (MLP 32) has a +8.43% surge in refusal mass, likely due to its position close to the output.
>
> The [linked figure](https://anonymous.4open.science/r/jetx-rebuttal-iclr-2026-E41D/toxic_refusal_mass_distribution.png) demonstrates the percentage change of toxic/refusal mass across different MLP paths. Positive change indicates increase of the toxic/refusal mass. If we denote M1 = mass from the augmented model, M2 = mass from the chat model, the percentage of change is calculated by (M1 - M2) / M2.
>
> Together, our results suggest that the further finetuned model might have acquired more knowledge of refusal phrases to reshape its toxicity behavior in "deeper" responses, rather than directly subtracting toxic knowledge mass. For us, this makes sense considering the finetuning process of [1], which involves augmenting the corpus to include examples with “Safety Recovery” in the middle of the responses. Our speculation is that the increased “refusal” mass emerges directly from this procedure, while the small reduction of toxic mass might only be an indirect result of the model updating its statistics to accommodate for the new data. This being said, the augmentation process does effectively change the model behaviour, although perhaps not by “directly removing” toxic knowledge.
>
> We hope this case on jet bigram mass satisfies your curiosity and demonstrates its practical utility.
>
> ## Q2 Cost of k>0 Lenses
> > Could the authors quantify the practical computational overhead of using the iterative jet lens (Fig. 9) versus the logit lens (Fig. 8)? How much more expensive is it, and does this limit its real-world applicability as a drop-in replacement?
>
> Thanks for the question. For each block, the k=1 iterative jet lens runtime cost is in the order of one forward pass plus one backward pass, O(F + B). Since iterative jet lenses have only one center, there is no need to optimize the jet weight w (which is the most costly computation). Therefore we believe that the iterative jet lens for k=1 is quite applicable and “interchangable” with the logit lens in real-world scenarios.
>
> ## Q3 Choice of k=1
> > Choice of k=1: The results for GPT-Neo are dramatically better for k=1, but the results for GPT-2-large (Figs. 11 vs. 12) seem comparable. Does this suggest k=1 is "good enough" for some model families? How should a practitioner choose the optimal k for a new model?
>
> Yes, k=1 is good enough in our empirical experiments. In general, we think modern language models exhibit mixed linear–nonlinear structure, where many features are linearly decodable [4–5]. In the interpretability literature, GPT-2 and Llama have cleaner residual geometry, so k=0 also works reasonably well. By contrast, GPT-Neo is well-known for its effective nonlinearity. We conjecture that these intrinsic model differences (that boil down to the specific choice of nonlinearities, training schemes, and so on) make k=0 either “succeed” or “fail”.
>
> At this moment, we do not have any “recipe” to choose an optimal k. We recommend searching k in {0, 1, 2}. Since the optimal order varies by architecture, light empirical tuning is generally the best approach. But we believe that studying this from a more theoretical perspective could surely be an interesting topic for further study.
>
> ## Q4 code releasing
> Yes, we will release the code.
>
> ## Q5 notation example 5
> Thanks for the sharp catch! It should be $J^1\gamma(x_1)(x=x_1 + x_2)$. We have corrected this typo.
>
> ## Q6 Lemma 1 Proof
> > Lemma 1 Proof: In the proof of Lemma 1 (Appendix A), the evaluation point for the jets appears to be mislabeled as 'x' when it should be 'y' (e.g., ). Can the authors confirm if this is a typo? Or clarify the difference between x and y?
>
> Thanks for the catch! It is a typo when we updated the notations. We have corrected it in the revision.
>
> ## Q7 "Data-Free" Clarification
> >The "data-free" claim for n-grams seems to be a source of confusion. The method is clearly input-dependent (it evaluates on token embeddings). Would "corpus-free" or "dataset-free" be a more accurate descriptor?
>
> Thanks for the suggestion. We agree with you and have changed it to "dataset-free". Note that the interesting bit about LLMs is the token embeddings while being the input are also part of the model themselves.

---

> > ### Author Response · Authors · 2025-11-26
> >
> > **References**
> >
> > [0] Elhoushi, Mostafa, et al. "Layerskip: Enabling early exit inference and self-speculative decoding." Proceedings of the 62nd Annual Meeting of the Association for Computational Linguistics (Volume 1: Long Papers). 2024.
> >
> > [1] Qi, Xiangyu, et al. "Safety Alignment Should be Made More Than Just a Few Tokens Deep." The Thirteenth International Conference on Learning Representations. arxiv, code, model weights of Llama2-7B-Chat-Augmented.
> >
> > [2] Jain, Neel, et al. "Refusal Tokens: A Simple Way to Calibrate Refusals in Large Language Models." Neurips Safe Generative AI Workshop 2024.
> >
> > [3] Arditi, Andy, et al. "Refusal in language models is mediated by a single direction." Advances in Neural Information Processing Systems 37 (2024): 136037–136083.
> >
> > [4] Park, Kiho, Yo Joong Choe, and Victor Veitch. "The Linear Representation Hypothesis and the Geometry of Large Language Models." International Conference on Machine Learning. PMLR, 2024.
> >
> > [5] Hernandez, Evan, et al. "Linearity of Relation Decoding in Transformer Language Models." The Twelfth International Conference on Learning Representations.
> >
> > [6] Hinton, Geoffrey E., et al. "Improving neural networks by preventing co-adaptation of feature detectors." arXiv preprint arXiv:1207.0580 (2012).
> >
> > [7] Srivastava, Nitish, et al. "Dropout: a simple way to prevent neural networks from overfitting." The journal of machine learning research 15.1 (2014): 1929–1958.
> >
> > [8] Gal, Yarin, and Zoubin Ghahramani. "Dropout as a bayesian approximation: Representing model uncertainty in deep learning." international conference on machine learning. PMLR, 2016.

---

### Author Response · Authors · 2025-11-26
**General Rebuttal Comment**

We extend our gratitude to all reviewers for the time, care, and insight invested in reading our work. Your comments have been invaluable in helping us refine this work.

We have uploaded a **revised PDF**, with **major updates highlighted in purple**. In this revision, we took the opportunity to slow down the exposition: clarifying and expanding the algorithms, smoothing transitions, and addressing each reviewer’s concerns. We also enriched the appendix with additional explanations and details to support the main body text.

We would like to invite the reviewers to **look over the individual responses and the revised manuscript**. We hope these changes bring better coherence, clarity, and transparency to the ideas we want to share.

---

### Author Response · Authors · 2025-12-04

We thank the reviewers and the area chair for their time evaluating this work. Here we summarize the revisions made during the rebuttal process. We introduced several updates to improve the manuscript:

* elaborated on remainders and the metrics to measure them;
* clarified the tractability of different jet expansions;
* added new experiments on RLHF toxicity/refusal behavior (see our response to F5cx);
* improved the connection between theory and empirical tools such as jet lenses and jet n-grams;
* expanded the appendix with clearer algorithmic descriptions;
* corrected notation and terminology throughout.

These updates, highlighted in the revised PDF in purple and explained in the rebuttal, aim to better present our main insight: that interpreting LLMs can be approached as finding a functional decomposition and carving out the part of interest -- allowing portions of the original LLM computation to be rewritten in a more inspectable form. Jet expansion is one way to concretize this perspective, and we hope it encourages the development of more diverse and scalable approaches. In general, we find this functional-decomposition viewpoint on interpretability beautiful and hope to share it with the community using this work. We appreciate your time and consideration.

---

### Meta-Review · Area_Chair_wb9f · 2026-01-08

**Summary:**

The paper introduces Jet Expansions, a framework using jet operators (generalizing Taylor series) to decompose LLMs into computational paths and remainders for better interpretability. It unifies tools like Logit Lens, extracts dataset-free n-grams, and analyzes toxicity in aligned models. Reviewers praised its novel theory, unification of existing methods, dataset-free approach, and empirical results on toxicity and n-grams. Concerns included remainder analysis, tractability for higher orders and large models, clarity on theory-empirical links, and novelty beyond reframing.

**Reviewer Concerns:**

Rebuttal addressed remainder sizes with cosine similarity metrics and clarifications on when small remainders are expected; clarified tractability by noting k=0/1 feasibility and exponential expansions as theoretical; improved exposition on n-grams and metrics; added experiments on RLHF toxicity/refusal. Outstanding: some lingering questions on optimal k choice and higher-order trustworthiness, but minor.

**Reviewer Scores:**

F5cx: likely unchanged at 8, as rebuttal aligned with strengths.
tf63: up to 7, with remainder concerns addressed.
KiGr: unchanged at 8, practicality clarified.
QKXz: up to 7 or 8, per their post-rebuttal comment.

---

### Decision · Program_Chairs · 2026-01-26

Accept (Poster)